# From Imagined Futures to Executable Actions: Mixture of Latent Actions for Robot Manipulation

Yajie Li [* 1 2]  Bozhou Zhang [* 1 2]  Chun Gu [1 2]  Zipei Ma [1 2]  Jiahui Zhang [1 2]  Jiankang Deng [3]  Xiatian Zhu [4]
Li Zhang [1 2]

https://logosroboticsgroup.github.io/MoLA

## Abstract

Video generation models offer a promising imagination mechanism for robot manipulation by predicting long-horizon future observations, but effectively exploiting these imagined futures for action execution remains challenging. Existing approaches either condition policies on predicted frames or directly decode generated videos into actions, both suffering from a mismatch between visual realism and control relevance. As a result, predicted observations emphasize perceptual fidelity rather than action-centric causes of state transitions, leading to indirect and unstable control. To address this gap, we propose **MoLA** (Mixture of Latent Actions), a control-oriented interface that transforms imagined future videos into executable representations. Instead of passing predicted frames directly to the policy, MoLA leverages a mixture of pretrained inverse dynamics models to infer a mixture of latent actions implied by generated visual transitions. These modality-aware inverse dynamics models capture complementary semantic, depth, and flow cues, providing a structured and physically grounded action representation that bridges video imagination and policy execution. We evaluate our approach on simulated benchmarks (LIBERO, CALVIN, and LIBERO-Plus) and real-world robot manipulation tasks, achieving consistent gains in task success, temporal consistency, and generalization.

*Equal contribution [1]School of Data Science, Fudan University, Shanghai, China [2]Shanghai Innovation Institute, Shanghai, China [3]Imperial College London, London, United Kingdom [4]University of Surrey, Guildford, United Kingdom. Correspondence to: Li Zhang <lizhangfd@fudan.edu.cn>.

*Proceedings of the 43rd International Conference on Machine Learning*, Seoul, South Korea. PMLR 306, 2026. Copyright 2026 by the author(s).

## 1. Introduction

Vision–Language–Action (VLA) models (Brohan et al., 2022; Zitkovich et al., 2023; O'Neill et al., 2024; Bjorck et al., 2025; Black et al., 2025; Cheang et al., 2024) have become a powerful paradigm for robot manipulation, learning to act directly from perception and language understanding, as shown in Figure 1 (a). By tightly coupling visual understanding with action prediction, VLA systems demonstrate strong generalization across tasks and settings. Alongside this direct perception-to-action route, another line of research explores imagination-based manipulation (Hu et al., 2025a; Liao et al., 2025; Feng et al., 2025b; Pai et al., 2025; Liang et al., 2025; Zhu et al., 2025; Chen et al., 2025a), where video generation models predict future observations to inform action selection, as shown in Figure 1 (b). Advances in video generation enable models to forecast plausible scene evolutions, capturing task semantics and object interactions beyond the current observation. Existing approaches typically follow one of two strategies: either using predicted frames as additional visual inputs to condition the policy (Hu et al., 2025a; Liao et al., 2025), or directly decoding generated futures into actions (Feng et al., 2025b; Chen et al., 2025a). The former leaves the burden of translating visual change into control to the action head, while the latter tightly couples execution to the accuracy of video prediction, making control highly sensitive to compounding prediction errors. As a result, effectively exploiting imagined futures for robot manipulation remains challenging.

A fundamental challenge is that visual predictions are not inherently action-oriented. Video generation models are optimized for perceptual realism rather than manipulation-relevant structure. As a result, even accurate future videos may fail to expose the physical causes of state transitions, making the mapping from imagined images to actions indirect and unstable. This gap motivates inverse-dynamics-style latent action models (Ye et al., 2025; Chen et al., 2025e; Bu et al., 2025b; Baker et al., 2022; Lee et al., 2024; Schmidt & Jiang, 2024; Bruce et al., 2024), which model the action responsible for a transition between observations.

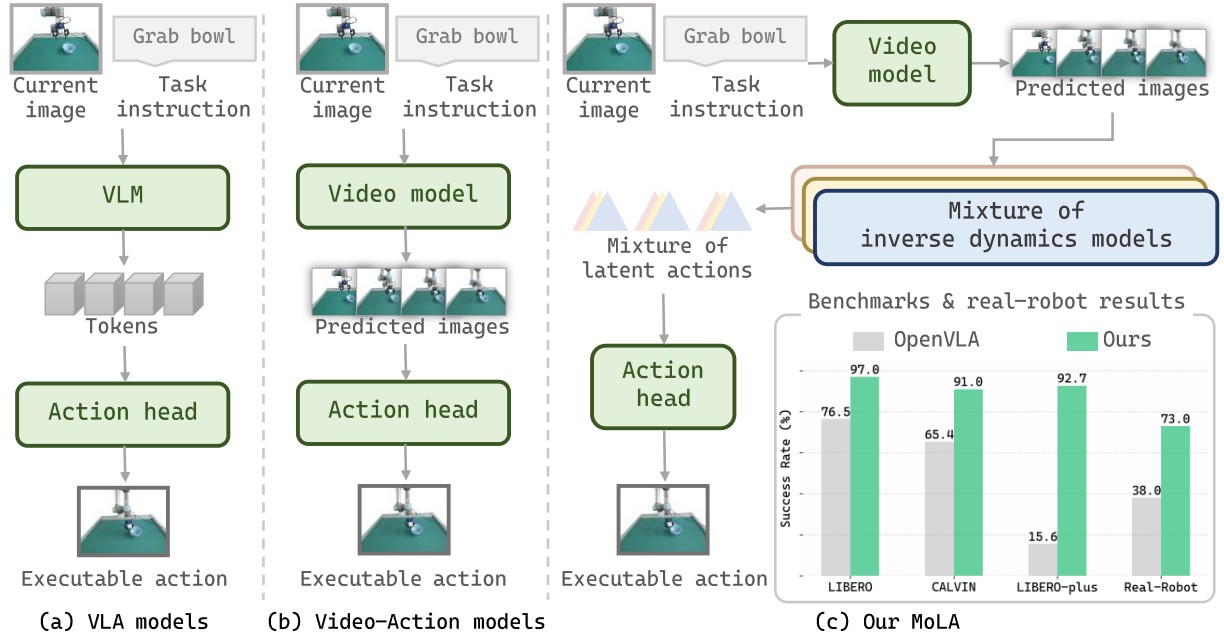

*Figure 1.* **Differences among (a) Vision–Language–Action models, (b) Video-Action models, and (c) our MoLA**. Our MoLA leverages a mixture of pretrained inverse dynamics models to infer a mixture of latent actions, which bridges video imagination and policy execution, enabling strong performance on both benchmarks and real-world robots.

Following works such as LAPA (Ye et al., 2025) and Uni-VLA (Bu et al., 2025b), we use IDM terminology in this broader latent-action sense: the model infers latent actions from observation transitions, and these latent actions are later decoded into continuous controls rather than treated as executable actions. By explicitly modeling the causal link from perception to action, inverse dynamics extracts manipulation-relevant information from visual change. This property suggests a simple but powerful idea: imagined future frames can be converted into executable representations by inferring the actions implied by predicted transitions. While inverse dynamics has been explored in robot learning, existing uses are typically task-specific or limited in scale, and do not provide (i) inverse dynamics models pretrained on large-scale robot datasets that can robustly operate on generated futures, nor (ii) a modality-aware set of inverse dynamics models that explicitly captures complementary semantic, geometric, and motion cues.

To make this connection robust, we pretrain inverse dynamics models on large-scale robot datasets (O'Neill et al., 2024; Bu et al., 2025a), enabling them to generalize across diverse manipulation behaviors and visual variations. Such pretraining allows the models to reliably infer actions not only from real observations but also from imperfect or generated future frames. Moreover, effective manipulation depends on complementary cues. Semantic changes convey task intent, depth captures geometric structure, and motion patterns reflect interaction dynamics. A single latent action

representation is often insufficient. We therefore train three modality-aware inverse dynamics models, specialized for semantic (Ravi et al., 2024), depth (Yang et al., 2024a), and flow-based motion (Karaev et al., 2025) information, and combine their outputs as a mixture of latent actions.

Based on this design, we propose **MoLA** (Mixture of Latent Actions), an imagination-based robot manipulation model whose core contribution is a latent-action interface between video generation and policy execution, as shown in Figure 1 (c). MoLA uses predicted future videos as an imagination space, converts them into a mixture of latent actions via pretrained inverse dynamics models, and conditions the policy on these action-centric representations. In doing so, MoLA enables reasoning over imagined futures directly in action space rather than image space.

We evaluate MoLA on standard simulated manipulation benchmarks, including LIBERO (Liu et al., 2023a), CALVIN (Mees et al., 2022), and LIBERO-Plus (Fei et al., 2025), and further validate its effectiveness on real-world robot manipulation tasks. Results show consistent improvements in success rate, temporal consistency, and generalization over methods that directly condition on predicted visual futures.

In summary, our main **contributions** are three-fold: **(i)** We introduce a latent-action interface for imagination-based robot manipulation, transforming predicted future videos into executable representations via inverse dynamics. **(ii)**

We propose MoLA, a manipulation model that couples video generation and action through a mixture of pretrained, modality-aware inverse dynamics models. **(iii)** We demonstrate the effectiveness of MoLA through extensive experiments on simulated benchmarks and real-world manipulation tasks.

## 2. Related Work

### 2.1. Vision–Language–Action Models

Vision–Language–Action (VLA) models aim to integrate visual perception, language understanding, and action generation to enable robots to perform diverse embodied tasks. Recent progress in large language models (Touvron et al., 2023; Brown et al., 2020; Roumeliotis & Tselikas, 2023; OpenAI, 2025), multimodal LLMs (OpenAI, 2023; Liu et al., 2023b; Qi et al., 2024), and large-scale robot datasets (O'Neill et al., 2024; Ebert et al., 2021; Khazatsky et al., 2024; Deng et al., 2025) has driven rapid advances in this area. Transformer-based architectures have emerged as the dominant paradigm for learning generalist robot policies from heterogeneous data, with Octo (Team et al., 2024) demonstrating strong generalization. In parallel, the RT series (Brohan et al., 2022; Zitkovich et al., 2023; Belkhale et al., 2024) pioneered fine-tuning multimodal language models on robot demonstrations, inspiring a wide range of subsequent VLA systems (Black et al., 2025; Kim et al., 2024; Li et al., 2024b; Liu et al., 2025a). Beyond direct action prediction, several works incorporate planning by generating goal images or future videos to guide policy execution (Du et al., 2024; Hu et al., 2025a; Black et al., 2023; Wu et al., 2024). Despite promising results, many VLA methods rely heavily on large-scale action-labeled robot data and often involve redundant visual reconstruction (Zheng et al., 2025b), limiting scalability and abstraction. To address these issues, emerging efforts explore learning from internet-scale, action-free videos, enabling more scalable policy learning without explicit action supervision.

### 2.2. Video-Action Models

Video-Action models build upon predictive world modeling by using video generation as a core mechanism to capture environment dynamics and guide robotic control. Recent advances in Transformer-based world models improve long-horizon visual prediction and scalability (Chen et al., 2022; Wang et al., 2024; Robine et al., 2023), enabling high-fidelity future observation synthesis. Several robotics works directly employ video generation models as policy backbones, decoding actions through inverse dynamics, tracking, or future-state matching (Black et al., 2023; Du et al., 2024; Hu et al., 2025a; Liang et al., 2024). Methods such as UniPi (Du et al., 2024), DREAMGEN (Jang et al., 2025), and joint co-generation frameworks (Guo et al., 2024; Zheng

et al., 2025b) demonstrate improved temporal consistency by coupling future video prediction with action generation. Overall, Video-Action models highlight the potential of predictive visual modeling to bridge perception, dynamics, and control within VLA systems.

### 2.3. Latent Action Models

Latent action models aim to reduce dependence on dense action annotations by learning compact representations that capture environment dynamics and agent behaviors. Early approaches learn variational latent spaces from action trajectories (Pu et al., 2016; Van Den Oord et al., 2017), while more recent methods such as VQ-BeT (Lee et al., 2024) and Quest (Mete et al., 2024) structure discrete action manifolds for behavior generation. To exploit unlabeled videos, several works infer latent actions from visual dynamics using inverse and forward models (Rybkin et al., 2019; Edwards et al., 2019; Bruce et al., 2024). Methods such as Genie (Bruce et al., 2024), LAPO (Schmidt & Jiang, 2024), and DynaMo (Cui et al., 2024) learn latent actions directly from visual observations without explicit action supervision, particularly for manipulation tasks. Recent VLA-oriented approaches further integrate latent actions into multimodal policies, enabling transfer from large-scale human videos (Ye et al., 2025; Chen et al., 2024). However, many latent action models rely on RGB reconstruction, which captures task-irrelevant appearance variations (Zhang et al., 2025a). To address this limitation, later works explore alternative supervision signals such as self-supervised visual features (Bu et al., 2025b; Chen et al., 2025e), object-centric representations (Yang et al., 2025d), and semantic grounding through language (Clark et al., 2025).

## 3. Method

### 3.1. Overview

MoLA (Mixture of Latent Actions) is an imagination-based manipulation framework that uses MoIDM as a latent-action interface to transform predicted future visual trajectories into structured, executable action representations, as shown in Figure 2. Given the current observation and task instruction, MoLA first employs a video generation model to synthesize future visual rollouts as an imagination space. These imagined futures are then processed by a mixture of modality-aware inverse dynamics models to infer complementary latent actions that explain the underlying visual transitions. The resulting mixture of latent actions is finally decoded into executable robot control commands by a diffusion-based action head.

In the following sections, we first introduce the video generation model used for future imagination (Section 3.2). We then describe the mixture of inverse dynamics models that

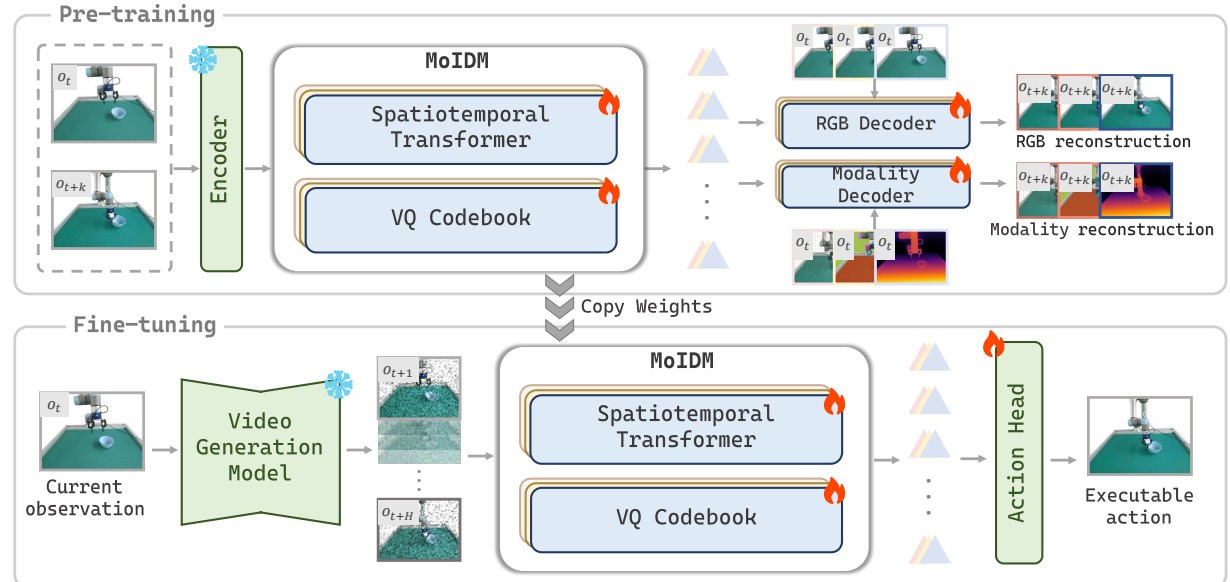

*Figure 2.* **Overview of MoLA.** Left: Pretraining of the mixture of inverse dynamics models (MoIDM). Modality-specific inverse dynamics models (e.g., flow-aware, semantic-aware, and depth-aware IDMs) are independently trained to extract and discretize fine-grained motion information from pairs of current and future frames. Right: End-to-end fine-tuning of the full model. The frozen video generation model is integrated with the pretrained MoIDM and the action head, while MoIDM and the action head are jointly optimized to generate executable action sequences.

extracts modality-specific latent actions (Section 3.3), followed by the action head that generates executable policies from the latent action mixture (Section 3.4). The overall training procedure is presented in Section 3.5.

### 3.2. Video Generation Model

Our MoLA adopts Stable Video Diffusion (SVD) (Blattmann et al., 2023) as the video generation backbone to provide visual imagination for robot manipulation. Given the current RGB observation $o_t^{\text{rgb}}$ and task instruction $l$, the model predicts a sequence of future frames $\hat{o}_{t:t+H}^{\text{rgb}}$ that captures the anticipated evolution of the scene under task execution.

Following the latent diffusion framework, SVD encodes video clips into a compact latent space and performs conditional denoising to generate future visual trajectories. The denoising process is conditioned on the latent embedding of the current observation and the instruction representation, enabling the model to synthesize task-consistent future rollouts.

During inference, MoLA restricts the generation process to a single denoising step for efficiency. Despite this simplification, the predicted latent futures retain sufficient visual and motion-relevant information to reflect meaningful scene dynamics for downstream action reasoning (Hu et al., 2025a). For multi-view robotic setups, future predictions are generated independently for each camera stream.

The video generation model serves as an imagination module that provides temporally coherent and visually informative future rollouts, which are subsequently transformed into action-centric representations by the inverse dynamics models.

### 3.3. Mixture of Inverse Dynamics Models (MoIDM)

Predicted future frames from the video generation model provide an imagination space, but they are not inherently action-centric. MoLA therefore introduces a mixture of modality-aware inverse dynamics models (MoIDM), which forms the core latent-action interface of the method by converting visual transitions into discrete latent actions between imagined futures and policy execution. We describe **(i)** the pretraining framework of each inverse dynamics model, and **(ii)** how MoIDM is integrated into MoLA for joint fine-tuning with the action head.

**Inverse Dynamics Pretraining.** Each inverse dynamics model adopts a modality-specific spatiotemporal Transformer, together with a modality-specific VQ codebook (Van Den Oord et al., 2017) for discrete latent actions, as shown in the left part of Figure 2. Given a current RGB frame $o_t^{\text{rgb}}$ and a future RGB frame $o_{t+k}^{\text{rgb}}$, we first extract RGB features using a ViT-based (Dosovitskiy, 2020) image encoder:

$$h_t^{\text{rgb}} = E_{\text{rgb}}(o_t^{\text{rgb}}), \qquad h_{t+k}^{\text{rgb}} = E_{\text{rgb}}(o_{t+k}^{\text{rgb}}). \qquad (1)$$

To model the temporal interaction between the current and

future observations, we introduce a set of learnable latent action queries, which are initialized and interact with the RGB features through a spatiotemporal Transformer:

$$\tilde{h}_{t \to t+k} = T^{(m)}\Big(q^{(m)}, h_t^{\text{rgb}}, h_{t+k}^{\text{rgb}}\Big), \qquad (2)$$

where $q^{(m)}$ denotes the latent action queries and $T^{(m)}$ denotes the modality-specific spatiotemporal Transformer that captures cross-frame correspondences.

The Transformer outputs are then mapped to discrete latent actions through a modality-specific vector-quantized codebook:

$$z_{t \to t+k}^{(m)} = \text{VQ}^{(m)}\Big(\tilde{h}_{t \to t+k}\Big), \qquad (3)$$

yielding compact action tokens that capture the causal transition between the two frames.

To induce modality awareness, each IDM is trained with distinct reconstruction targets. The predicted latent action is combined with the current RGB feature and decoded by a shared ViT-based RGB decoder to reconstruct the future RGB frame:

$$\hat{o}_{t+k}^{\text{rgb}} = D_{\text{rgb}}\Big(h_t^{\text{rgb}}, z_{t \to t+k}^{(m)}\Big), \qquad (4)$$

which is optimized for *all* inverse dynamics models.

In addition, modality-specific supervision is provided using frozen foundation models. We employ Depth Anything v2 (Yang et al., 2024a) to extract depth features and depth ground-truth maps, SAM2 (Ravi et al., 2024) to extract semantic features and segmentation targets, and Co-Tracker3 (Karaev et al., 2025) to extract motion flow features and motion flow targets. Let

$$h_t^{(m)} = F^{(m)}(o_t^{\text{rgb}}), \qquad g_{t+k}^{(m)} = F^{(m)}(o_{t+k}^{\text{rgb}}), \qquad (5)$$

where $F^{(m)}$ denotes the corresponding foundation encoder and $g_{t+k}^{(m)}$ represents the modality-specific future ground-truth.

A ViT-based modality decoder then reconstructs the future modality representation:

$$\hat{g}_{t+k}^{(m)} = D^{(m)}\Big(h_t^{(m)}, z_{t \to t+k}^{(m)}\Big). \qquad (6)$$

Each inverse dynamics model is pretrained independently with its own modality reconstruction objective, resulting in three specialized IDMs that share the same RGB-based action inference pipeline while becoming sensitive to semantic, geometric, or motion cues through distinct supervision.

**Integration within MoLA (Joint Fine-Tuning).** In the MoLA framework, the video generation model is frozen and

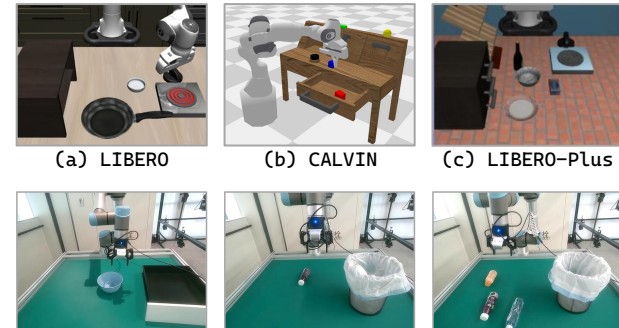

(a) LIBERO  (b) CALVIN  (c) LIBERO-Plus

(d) Real-world UR5e robot

*Figure 3.* Experiments are conducted on the CALVIN ABC-D, LIBERO, and LIBERO-Plus benchmarks, as well as on a real-world UR5e robot. We evaluate MoLA across three simulated benchmarks and the real-world setup.

used to generate imagined future RGB rollouts $\hat{o}_{t:t+H}^{\text{rgb}}$. For the predicted future frame $\hat{o}_{t+k}^{\text{rgb}}$, MoIDM infers modality-specific discrete latent actions through the spatiotemporal Transformer and modality-specific VQ codebooks, producing a mixture of latent actions:

$$\mathcal{Z}_{t \to t+k} = \Big\{ z_{t \to t+k}^{(\text{sem})}, \ z_{t \to t+k}^{(\text{depth})}, \ z_{t \to t+k}^{(\text{flow})} \Big\}. \qquad (7)$$

All inferred latent actions, together with the corresponding generated visual features, are provided as conditioning inputs to the action head. During downstream training, the parameters of MoIDM and the action head are jointly optimized end-to-end, allowing the latent actions to adapt to the characteristics of imagined futures and the control objective, while keeping the video generation model fixed, as shown in the right part of Figure 2.

### 3.4. Action Head

We employ the Diffusion Transformer (Peebles & Xie, 2023; Reuss et al., 2024) architecture as the action decoding module, which is trained from scratch to generate executable control policies. The action head takes as input the mixture of latent action representations extracted from multiple pretrained inverse dynamics models, together with visual features of the predicted future frames generated by the video model, and predicts action sequences in a conditional generative manner. Instead of standard diffusion training, the model is optimized using a flow matching objective (Lipman et al., 2023), which learns a continuous transformation from noisy action samples to clean target actions under the given conditional features. Through this process, the policy effectively captures complex and multimodal action distributions while enabling efficient and stable training.

## 3.5. Model Training

We train MoLA in three stages to progressively align video imagination, latent action inference, and policy execution.

**Stage I: Video Generation Model Fine-Tuning.** We first fine-tune the video generation model on large-scale robot manipulation datasets to adapt it to task-specific visual dynamics and viewpoints. The training datasets and configurations are summarized in Appendix Table 6.

**Stage II: MoIDM Pretraining.** Next, we pretrain the mixture of inverse dynamics models using similar robot datasets as in Stage I.

**Stage III: End-to-End Fine-Tuning.** Finally, we integrate the frozen video generation model with MoIDM and the diffusion-based action head. The MoIDM and action head are jointly fine-tuned end-to-end using downstream manipulation demonstrations.

# 4. Experiments

In this section, we perform extensive experiments in both simulated and real-world settings to evaluate the effectiveness of our MoLA, as illustrated in Figure 3.

## 4.1. Implementation Details

Model and training details, detailed ablation tables, and qualitative analysis are provided in Appendix Sections A.1, A.2, and A.6.

## 4.2. Simulation Benchmark Experiments

### 4.2.1. BENCHMARKS

We evaluate our approach on three widely used robotic manipulation benchmarks: CALVIN (Mees et al., 2022), LIBERO (Liu et al., 2023a), and LIBERO-Plus (Fei et al., 2025). CALVIN is a long-horizon, language-conditioned manipulation benchmark with multi-sensor demonstrations across four simulated environments; we follow the challenging ABC-D split, training on environments A–C and evaluating on the unseen environment D. LIBERO consists of four lifelong learning task suites covering spatial reasoning, object generalization, goal adaptation, and long-horizon planning; we pretrain on LIBERO-90 and fine-tune and evaluate on each suite. LIBERO-Plus further extends LIBERO with controlled perturbations across seven dimensions to assess robustness, comprising 10,030 tasks.

### 4.2.2. RESULTS AND ANALYSIS

We conduct systematic comparisons on the CALVIN and LIBERO. The experimental results clearly demonstrate the strong performance of MoLA: our method achieves the highest average success rate across all three benchmarks and exhibits superior performance on all tasks compared to existing approaches.

The quantitative results on the CALVIN benchmark are reported in Table 1. Models marked with * indicate results quoted from previous work (Zhang et al., 2026), following the standard evaluation protocol. As shown in Table 2, MoLA achieves strong gains on all four LIBERO task suites. Representative failure cases and their visualizations are provided in Appendix Section A.6.

Furthermore, as shown in Table 3, on the large-scale **LIBERO-Plus** benchmark comprising **10,030 tasks**, MoLA achieves state-of-the-art performance across all task suites. Specifically, it outperforms the strongest baseline, OpenVLA-OFT+, by an average success rate margin of **13.2%**, demonstrating a clear absolute performance advantage. These results indicate that MoLA possesses stronger multi-task learning capabilities and superior generalization performance in simulated environments.

### 4.2.3. DATA EFFICIENCY

Acquiring large-scale robot interaction data is both expensive and time-consuming, motivating learning approaches that can effectively leverage limited supervision. To evaluate the data efficiency of our method, we benchmark it on the CALVIN ABC-D benchmark by fine-tuning pretrained policies using varying fractions of the training set (10%, 20%, 50%, and 100%). As shown in Figure 4, our method consistently outperforms VPP across all data regimes. The advantage is particularly pronounced in low-data settings, and these results indicate a stronger ability to leverage pretrained representations under limited supervision. Our method continues to achieve superior performance at 50% and 100% of the training data, demonstrating both data efficiency and scalability.

## 4.3. Real-World Experiments

### 4.3.1. EXPERIMENT SETUP AND BASELINES

For real-world experiments, we use a Universal Robots UR5e robotic arm. The setup includes a RealSense D455 camera providing third-person observations and a RealSense D405 camera mounted on the gripper to capture egocentric views. We collect 1,000 demonstration trajectories across five manipulation tasks, such as placing a bottle into a bin and grasping a bowl. During evaluation, each task is performed for 50 trials, with objects randomly initialized on the table at the beginning of each trial. In each trial, the robot is allowed up to 20 attempts to complete the task. A trial is considered successful if the task is completed within the allowed attempts; otherwise, it is counted as a failure.

*Table 1.* **Evaluation results on the CALVIN ABC-D benchmark.** MoLA achieves strong performance on all tasks, demonstrating effective generalization. The best results are **bolded**. * indicates results reproduced by us or reported in previous work.

| Methods | Task completed in a row | | | | | Avg. Len. ↑ |
|---|---|---|---|---|---|---|
| | 1 | 2 | 3 | 4 | 5 | |
| 3D Diffusor Actor (Ke et al., 2024) | 92.2 | 78.7 | 63.9 | 51.2 | 41.2 | 3.27 |
| OpenVLA (Kim et al., 2024) | 91.3 | 77.8 | 62.0 | 52.1 | 43.5 | 3.27 |
| UniVLA (Bu et al., 2025b) | 95.5 | 85.8 | 75.4 | 66.9 | 56.5 | 3.80 |
| $\pi_0*$ (Black et al., 2025) | 93.8 | 85.0 | 76.7 | 68.1 | 59.9 | 3.84 |
| $\pi_{0.5}*$ (Intelligence et al., 2025) | 94.8 | 87.4 | 78.2 | 71.7 | 64.3 | 3.97 |
| GR00T N1* (Bjorck et al., 2025) | 94.2 | 86.1 | 79.6 | 73.9 | 66.8 | 4.01 |
| CLOVER (Bu et al., 2024b) | 96.0 | 83.5 | 70.8 | 57.5 | 45.4 | 3.53 |
| UP-VLA (Zhang et al., 2025c) | 92.8 | 86.5 | 81.5 | 76.9 | 69.9 | 4.08 |
| Seer (Tian et al., 2025) | 96.3 | 91.6 | 86.1 | 80.3 | 74.0 | 4.28 |
| VPP (Hu et al., 2025a) | 96.5 | 90.9 | 86.6 | 82.0 | 76.9 | 4.33 |
| DreamVLA (Zhang et al., 2025e) | 98.2 | 94.6 | 89.5 | 83.4 | 78.1 | 4.44 |
| MoLA (Ours) | **98.5** | **95.0** | **91.1** | **88.1** | **82.6** | **4.55** |

*Table 2.* **Evaluation results on the LIBERO benchmark.** MoLA achieves the best performance, surpassing previous approaches on all four task suites. References: Octo (Team et al., 2024), OpenVLA (Kim et al., 2024), SpatialVLA (Qu et al., 2025), CoT-VLA (Zhao et al., 2025a), VPP (Hu et al., 2025a). The best results are **bolded**. * indicates results reproduced by us.

| Methods | Scores (%) | | | | Avg. ↑ |
|---|---|---|---|---|---|
| | Spa. | Obj. | Goal | Long | |
| Octo | 78.9 | 85.7 | 84.6 | 51.1 | 75.1 |
| OpenVLA | 84.7 | 88.4 | 79.2 | 53.7 | 76.5 |
| SpatialVLA | 88.2 | 89.9 | 78.6 | 55.5 | 78.1 |
| CoT-VLA | 87.5 | 91.6 | 87.6 | 69.0 | 83.9 |
| VPP* | 85.0 | 95.0 | 92.0 | 91.5 | 90.9 |
| MoLA (Ours) | **93.0** | **99.5** | **99.5** | **96.0** | **97.0** |

*Table 3.* **Evaluation results on the LIBERO-Plus benchmark.** MoLA surpasses the strongest baseline, OpenVLA-OFT+, by an average margin of 13.2 percentage points. References: OpenVLA (Kim et al., 2024), WorldVLA (Cen et al., 2025b), NORA (Hung et al., 2025), UniVLA (Bu et al., 2025b), $\pi_0$ (Black et al., 2025), $\pi_0$-Fast (Pertsch et al., 2025), RIPT-VLA (Tan et al., 2025b), OpenVLA-OFT (Kim et al., 2025), OpenVLA-OFT+ (Fei et al., 2025). Avg. denotes the overall task score and the best results are **bolded**.

| Methods | Scores (%) | | | | Avg. ↑ |
|---|---|---|---|---|---|
| | Spa. | Obj. | Goal | Long | |
| OpenVLA | 19.4 | 14.0 | 15.1 | 14.3 | 15.6 |
| WorldVLA | 32.5 | 28.6 | 31.8 | 8.2 | 25.0 |
| NORA | 47.6 | 34.4 | 38.8 | 36.3 | 39.0 |
| UniVLA | 55.5 | 36.7 | 40.7 | 39.9 | 42.9 |
| $\pi_0$ | 60.7 | 61.4 | 44.9 | 48.4 | 53.6 |
| $\pi_0$-Fast | 74.4 | 72.7 | 57.5 | 43.4 | 61.6 |
| RIPT-VLA | 85.8 | 64.3 | 58.0 | 67.5 | 68.4 |
| OpenVLA-OFT | 84.0 | 66.5 | 63.0 | 66.4 | 69.6 |
| OpenVLA-OFT+ | 86.1 | 84.5 | 70.7 | 77.7 | 79.5 |
| MoLA (Ours) | **97.5** | **96.3** | **85.1** | **91.8** | **92.7** |

We evaluate both the baseline methods and our MoLA under two experimental settings, as shown in Appendix Figure 13. The first setting is in-distribution, where evaluation is conducted in environments consistent with the training scenarios to assess the model's learning capability. The second setting is out-of-distribution, which aims to evaluate generalization performance and instruction-following ability. For the out-of-distribution setting, we introduce environmental variations, including distracting objects and changes in lighting conditions.

We compare our method with a policy learning model, Diffusion Policy (Chi et al., 2023), VLA models including OpenVLA (Kim et al., 2024) and $\pi_{0.5}$ (Intelligence et al., 2025), as well as a Video-Action model, VPP (Hu et al., 2025a). For a fair comparison, all methods are fine-tuned on the same collected demonstration dataset.

### 4.3.2. RESULTS AND ANALYSIS

We conduct real-world experimental comparisons between our MoLA and several baseline models, as shown in Table 4. Our method significantly outperforms the policy learning model Diffusion Policy (Chi et al., 2023) and the VLA

model OpenVLA (Kim et al., 2024), and also demonstrates clear gains over the more directly comparable world-model-based baseline VPP (Hu et al., 2025a). The company-scale end-to-end VLA $\pi_{0.5}$ (Intelligence et al., 2025) achieves the highest average score in this comparison, and we include it as a strong reference point rather than as a strict apples-to-apples comparator to the world-model-plus-interface setting studied here. Within that setting, the results highlight the effectiveness of the proposed mixture of latent actions in bridging video imagination and policy execution.

### 4.4. Ablation Study

In this section, we perform a comprehensive ablation study to analyze the key design choices of MoLA.

**Q1: How Does Each IDM Contribute to Performance?**

MoLA converts imagined future videos into action-centric

*Table 4.* **Evaluation results on the real-world robot.** We evaluate MoLA and the baseline methods under both in-distribution and out-of-distribution settings. Comparison among three model categories: policy learning methods, VLA models, and Video-Action models.

| Methods | In-distribution | | Out-of-distribution | | Avg. ↑ |
|---|---|---|---|---|---|
| | Place bottle | Grab bowl | Distracting objects | Lighting changes | |
| Diffusion Policy (Chi et al., 2023) | 44.0% | 56.0% | 28.0% | 38.0% | 41.5% |
| OpenVLA (Kim et al., 2024) | 52.0% | 46.0% | 24.0% | 30.0% | 38.0% |
| $\pi_{0.5}$ (Intelligence et al., 2025) | 82.0% | 90.0% | 64.0% | 74.0% | 77.5% |
| VPP (Hu et al., 2025a) | 66.0% | 72.0% | 52.0% | 58.0% | 62.0% |
| MoLA (Ours) | **76.0%** | **92.0%** | **60.0%** | **64.0%** | **73.0%** |

*Table 5.* **Effect of MoIDM on CALVIN ABC-D.** We progressively add different IDMs while keeping all other components fixed. The baseline aggregates predicted future visual features with a temporal Transformer and feeds them directly to the action head, without MoIDM or a VQ-VAE-style discrete latent-action bottleneck.

| Combination | Flow | Depth | Sem. | Task completed in a row | | | | | Avg. Len. ↑ |
|---|---|---|---|---|---|---|---|---|---|
| | | | | 1 | 2 | 3 | 4 | 5 | |
| Baseline | ✗ | ✗ | ✗ | 95.9 | 90.4 | 85.1 | 80.0 | 72.8 | 4.24 |
| Sem. only | ✗ | ✗ | ✓ | 95.4 | 90.7 | 86.1 | 82.2 | 76.8 | 4.31 |
| Depth only | ✗ | ✓ | ✗ | 96.4 | 91.3 | 87.0 | 82.5 | 77.4 | 4.35 |
| Flow only | ✓ | ✗ | ✗ | 95.6 | 91.7 | 88.2 | 84.6 | 79.2 | 4.39 |
| Flow + Depth | ✓ | ✓ | ✗ | 97.1 | 93.1 | 89.3 | 85.5 | 80.9 | 4.46 |
| All modalities | ✓ | ✓ | ✓ | **98.5** | **95.0** | **91.1** | **88.1** | **82.6** | **4.55** |

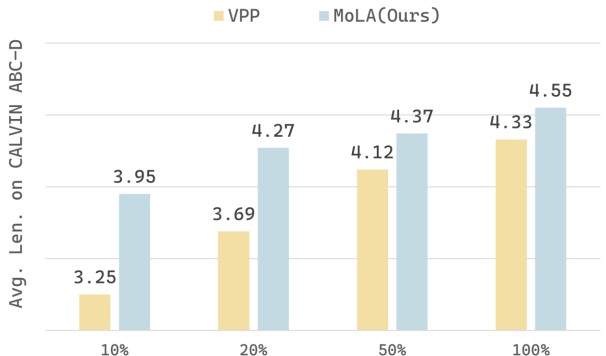

*Figure 4.* **Data efficiency comparison on CALVIN ABC-D.** Performance of pretrained policies fine-tuned using different fractions of the training data.

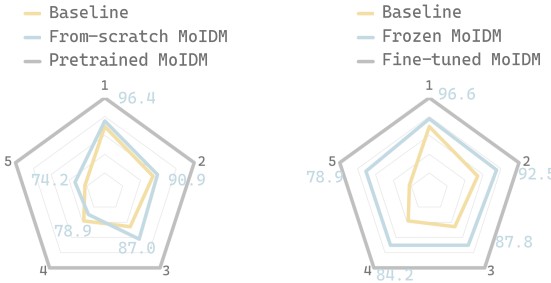

*(a)* Effect of MoIDM pretraining.

*(b)* Effect of MoIDM fine-tuning.

*Figure 5.* **Ablation studies on MoIDM.** These experiments illustrate the impact of pretraining and fine-tuning strategies on model performance.

representations via MoIDM. To isolate the contribution of each modality, we progressively add different IDMs while keeping all other components fixed. Here, the baseline aggregates future visual features predicted by the video model with a temporal Transformer and feeds the resulting representation directly to the action head. As shown in Table 5, among single-modality variants, the flow-aware IDM yields the largest improvement over directly conditioning the policy on predicted frames, underscoring the importance of motion cues for modeling interaction dynamics. Adding the depth-aware IDM further boosts performance, since it encodes three-dimensional spatial structure. Finally, incorporating the semantic-aware IDM achieves the best overall results by providing high-level object and task context. The

weaker baseline also suggests that without IDM, the action head must learn the observation-to-action mapping directly from aggregated future visual features, which substantially increases the difficulty of control modeling. Performance improves monotonically as modalities are added, indicating that the three IDMs capture complementary aspects of imagined futures; additional comparisons against single-branch alternatives are provided in Appendix Table 13.

**Q2: What Is the Role of MoIDM Pretraining?**

MoIDM in MoLA is pretrained on large-scale robot manipulation datasets. We assess the importance of pretraining by comparing against MoIDM trained from scratch under identical downstream settings. As shown in Figure 5a, removing pretraining results in substantial performance degradation

across all benchmarks. MoIDM trained from scratch struggles to infer actions from generated future frames, resulting in uninformative latent actions that weakly condition the policy.

### Q3: Should MoIDM Be Frozen During Policy Fine-Tuning?

We further compare freezing pretrained MoIDM with joint fine-tuning during downstream training. As shown in Figure 5b, freezing consistently underperforms joint optimization. While freezing preserves general representations, frozen MoIDM cannot adapt to the evolving policy, leading to misaligned latent actions. End-to-end fine-tuning allows MoIDM to co-adapt with both components, yielding task-aligned and physically grounded action representations.

### Q4: What Action Head Best Translates Latent Actions into Executable Controls?

The discrete latent actions inferred by MoIDM serve as intermediate cues distilled from imagined futures rather than final controls. The policy must still model a continuous, temporally correlated, and often multimodal action distribution. Appendix Table 11 compares the default DiT-based action head trained with flow matching against a lighter autoregressive Transformer over discretized action tokens. The autoregressive variant remains effective, confirming that the latent actions are informative, but it is consistently weaker than the DiT-based head, which better captures long-horizon dependencies, multimodality, and stable control under imperfect latent cues.

### Q5: Can MoLA Benefit from a Stronger Video Generation Backbone?

MoLA is not tied to Stable Video Diffusion and can naturally adopt stronger DiT-based video generators. Appendix Table 12 reports results obtained by replacing SVD with Wan2.2 5B (Wan et al., 2025) as the imagination backbone on CALVIN ABC-D. The improvement over the default backbone shows that MoLA is compatible with stronger video generators and can benefit from advances in the underlying imagination model.

### Q6: Is the Proposed Multi-IDM Decomposition Necessary?

Appendix Table 13 compares the proposed three-IDM decomposition against several unified alternatives. The direct multimodal-input baseline feeds depth, semantic, and motion flow features directly to the action head without an explicit inverse-dynamics-style bottleneck. Its weaker performance indicates that latent-action compression is the principal mechanism that connects imagined futures with executable control, while structured multimodal signals make this interface more physically grounded by encouraging semantic, geometric, and motion-aware decomposition.

VGGT primarily provides geometric and spatial priors without explicit functional decomposition. A single RGB IDM entangles multiple factors and is more sensitive to control-irrelevant appearance variation. A shared multi-modal IDM forces all signals into a single codebook and latent space, which increases objective interference and representation trade-offs. In contrast, MoLA employs three specialized IDMs to preserve complementary inductive biases that the action head can fuse into more robust and interpretable action-centric latent representations.

### Q7: Are Explicit Imagined Futures Necessary at Inference Time?

MoIDM itself does not predict future observations. Instead, it takes the current frame and an imagined future frame and extracts action-centric latent actions from the transition between them. Appendix Table 14 isolates the role of explicit online imagined futures through two inference-time controls: feeding MoIDM either two identical current frames or the current frame together with a noisy version of itself. Both settings preserve MoIDM while removing the task-conditioned future transition cues supplied by the world model. The resulting performance degradation indicates that MoLA benefits from both latent action learning and explicit future cues during inference.

## 5. Conclusion

In this paper, we proposed MoLA, a latent-action interface within video-based imagination for robot manipulation, implemented through a mixture of pretrained, modality-aware inverse dynamics models. By converting predicted future observations into structured latent action representations, MoLA enables reasoning over imagined trajectories directly in action space, alleviating the mismatch between visual realism and control relevance. Extensive experiments on simulation benchmarks and real-world robotic tasks demonstrate consistent improvements in task success and generalization over strong baselines. These results highlight the effectiveness of leveraging complementary semantic, geometric, and motion cues for action-centric future reasoning.

## Impact Statement

This paper presents work whose goal is to advance machine learning for robot manipulation. Potential societal consequences include both beneficial applications, such as safer and more efficient assistive or industrial robots, and risks associated with deploying learned policies in physical environments. We emphasize that real-world deployment should include appropriate safety validation, human oversight, and consideration of domain-specific ethical and societal constraints.

## Acknowledgments

This work was supported in part by New Generation Artificial Intelligence-National Science and Technology Major Project (2025ZD0123004), Ningbo grant (2025Z038) and National Natural Science Foundation of China (Grant No. 62376060).

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

# A. Appendix

## A.1. Training, Inference, and Model Details

*Table 6.* **Dataset statistics in the video generation model pretraining.** Number of trajectories used from each dataset during video generation model pretraining.

| Training dataset mixture | Trajectory numbers |
|---|---|
| Something-something-v2 (Goyal et al., 2017) | 170,962 |
| RT-1 (Brohan et al., 2022) | 84,236 |
| Bridge (Ebert et al., 2021) | 27,196 |
| CALVIN-ABC (Mees et al., 2022) | 19,120 |
| LIBERO (Liu et al., 2023a) | 6,500 |
| LIBERO-Plus (Fei et al., 2025) | 14,347 |

**Video Generation Model Pretraining Details.** During the pretraining of the video generation model, we used a mixture of human manipulation datasets and robotic video datasets, including Something-something-v2 (Goyal et al., 2017), RT-1 (Brohan et al., 2022), Bridge (Ebert et al., 2021), CALVIN (Mees et al., 2022), LIBERO (Liu et al., 2023a), and LIBERO-Plus (Fei et al., 2025). To balance the contribution of each dataset during training, we sample trajectories proportionally to ensure diverse coverage of both human and robotic interactions. This strategy helps the model generalize better across different types of video content. Detailed dataset statistics and sampling configurations are summarized in Table 6.

**MoIDM Pretraining Details.** MoIDM is built by independently training modality-specific IDMs, including depth-aware, flow-aware, and semantic-aware variants. During standalone training on each benchmark dataset, each IDM uses decoders to reconstruct both its corresponding modality-specific representation (e.g., depth maps, semantic segmentation, or optical flow) and RGB frames, thereby enforcing motion consistency across modalities. When incorporated into the full model, the decoders are removed, retaining only the spatiotemporal Transformer and the VQ codebook.

**MoLA Fine-Tuning Details.** During the MoLA fine-tuning phase, the video generation model is frozen, and only MoIDM and the action head are jointly fine-tuned to adapt to downstream control tasks. The model and training parameters for all components of MoLA are summarized in Table 7.

**Inference Time and Model Size.** As shown in Table 8, we evaluate the module-level inference time of MoLA on the CALVIN (Mees et al., 2022) benchmark and report the model size. This measurement corresponds to the same model configuration used throughout the paper. All experiments in Table 8 are conducted on an NVIDIA GeForce RTX 4090 GPU, with each component's timing averaged over five runs.

**Real-World Inference Cost.** The remaining experiments use the same model configuration and $256 \times 256$ visual inputs, leading to highly similar inference cost. To make deployment overhead more explicit, Table 9 reports module-level latency in the real-world setting. All timing results in this table are measured on an NVIDIA GeForce RTX 4090 GPU and averaged over ten runs.

**Effect of Video Diffusion Inference Steps.** Table 10 studies the effect of the number of video diffusion denoising steps at test time. One-step denoising already provides highly effective control-relevant signals while offering by far the lowest inference cost. Increasing the number of denoising steps does not yield a monotonic performance gain, indicating that MoLA does not require photorealistic future video, but rather compact future hypotheses that preserve the temporal structure most useful for action inference. All timing results in Table 10 are measured on an NVIDIA GeForce RTX 4090 GPU and averaged over ten runs.

*Table 7.* **Model and training parameters** of MoLA.

**(a) Model Parameters**

| Module | Name | Value |
|---|---|---|
| Video generation model | Type | Stable Video Diffusion |
| | Language shape | $20 \times 512$ |
| | Image shape | $256 \times 256$ |
| Spatiotemporal Transformer | Num queries | 8 |
| | Num layers | 4 |
| | Hidden size | 768 |
| | Num heads | 12 |
| VQ codebook | Num codes | 128 |
| | Latent dim | 32 |
| Action head | Type | Diffusion Transformer |
| | Latent dim | 384 |
| | Condition shape | $225 \times 384$ |
| | Num heads | 8 |
| | Encoder layers | 4 |
| | Decoder layers | 4 |
| | Sampling steps | 10 |

**(b) Training Parameters**

| Module | Name | Value |
|---|---|---|
| Video generation model | GPU | NVIDIA H100 |
| | Number of GPUs | 8 |
| | Pretraining time | 5 days |
| | Num frames | 16 |
| | Inference steps | 30 |
| | Batch size | 6 |
| | Optimizer | AdamW |
| | Learning rate | $5 \times 10^{-6}$ |
| | Weight decay | 0 |
| | Training steps | 500,000 |
| Mixture of inverse dynamics models | GPU | NVIDIA H100 |
| | Number of GPUs | 8 |
| | Pretraining time | 16 hours |
| | Batch size | 256 |
| | Optimizer | AdamW |
| | Lr max | $1 \times 10^{-4}$ |
| | Lr schedule | cosine decay |
| | Weight decay | $1 \times 10^{-4}$ |
| | Warmup steps | 1000 |
| | Epoch | 20 |
| MoLA | GPU | NVIDIA H100 |
| | Number of GPUs | 8 |
| | Fine-tuning time | 20 hours |
| | Batch size | 32 |
| | Optimizer | AdamW |
| | Learning rate | $1 \times 10^{-4}$ |
| | Weight decay | $5 \times 10^{-2}$ |
| | Epoch | 20 |

## A.2. Additional Ablations

## A.3. Limitations and Future Work

While MoLA effectively bridges imagined futures and executable actions through latent action representations, it does not explicitly incorporate high-level semantic reasoning or task understanding. Without large language models, the system

*Table 8.* **The inference time and model size of our MoLA.**

| Model part | Inference time | Model size |
|---|---|---|
| Video generation model | 79.85 ms | 1.53 B |
| Mixture of inverse dynamics models | 12.93 ms | 90.58 M |
| Action head | 8.26 ms | 22.90 M |

*Table 9.* **Module-level inference time in the real-world setting.**

| Model part | Real-world inference time |
|---|---|
| Video generation model | 81.23 ms |
| Mixture of inverse dynamics models | 11.05 ms |
| Action head | 9.62 ms |

*Table 10.* **Effect of the number of video diffusion denoising steps at test time.**

| Number of video diffusion inference steps | Avg. Len. ↑ | Total inference time |
|---|---|---|
| 1 | 4.55 | 0.1296 s |
| 2 | 4.52 | 0.2431 s |
| 10 | **4.56** | 1.1659 s |
| 20 | 4.49 | 2.3185 s |

*Table 11.* **Comparison of action heads on CALVIN ABC-D.** The latent actions remain the same, while only the final action generation head is changed. The best results are **bolded**.

| Action head | Task completed in a row | | | | | Avg. Len. ↑ |
|---|---|---|---|---|---|---|
| | 1 | 2 | 3 | 4 | 5 | |
| AR token Transformer | 95.7 | 91.4 | 88.6 | 85.1 | 78.9 | 4.40 |
| DiT + flow matching | **98.5** | **95.0** | **91.1** | **88.1** | **82.6** | **4.55** |

remains limited in its ability to interpret complex instructions, reason over long-horizon goals, and adapt to novel scenarios beyond visual dynamics. In future work, we plan to integrate MoLA with large language models to enable stronger semantic grounding and decision-making. Moreover, leveraging large-scale human demonstration data and internet-scale videos offers a promising direction to further enhance generalization and scalability for real-world robotic manipulation.

### A.4. A Thorough Literature Review

**Vision–Language–Action Models**   Vision–Language–Action (VLA) models aim to unify visual perception, language understanding, and action generation to enable robots to perform diverse embodied tasks. With the rapid progress of large language models (LLMs) (Touvron et al., 2023; Brown et al., 2020; Roumeliotis & Tselikas, 2023; OpenAI, 2025), multimodal LLMs (OpenAI, 2023; Liu et al., 2023b; Dong et al., 2024; Qi et al., 2024; Peng et al., 2025), and the availability of large-scale robot datasets (O'Neill et al., 2024; Ebert et al., 2021; Khazatsky et al., 2024; Deng et al., 2025), VLAs have become a central research direction in robot learning. Recent works leverage Transformer-based architectures to learn generalist robot policies from large-scale heterogeneous data, with Octo (Team et al., 2024) demonstrating strong generalization capabilities. In contrast, the RT series (Brohan et al., 2022; Zitkovich et al., 2023; Belkhale et al., 2024) pioneers the fine-tuning of multimodal large language models on robot demonstrations, achieving strong performance in both accuracy and generalization, and inspiring numerous subsequent improvements (Li et al., 2023; Black et al., 2025; Team et al., 2024; Kim et al., 2024; Li et al., 2024a; Lin et al., 2025; Zhang et al., 2024a; Wen et al., 2025b; Qu et al., 2025; Li et al., 2024b; Liu et al., 2025a; Intelligence et al., 2025; Bu et al., 2025a; Reuss et al., 2025; Song et al., 2025; Li et al., 2023; Wu et al., 2024; Jia et al., 2025a; Qi et al., 2025).

Beyond direct action prediction, some methods incorporate planning mechanisms through goal images or future video generation to guide policy execution (Du et al., 2024; Zawalski et al., 2024; Zhen et al., 2024a; Nasiriany et al., 2024; Gu et al., 2023; Zhang et al., 2024b; Wen et al., 2024a; Hu et al., 2025a; Black et al., 2023; Wu et al., 2024; Bu et al., 2024b). Despite promising results, these approaches often suffer from redundant visual reconstruction (Zheng et al., 2025b) and

*Table 12.* **Comparison of video generation backbones on CALVIN ABC-D.** MoLA is compatible with stronger imagination backbones and benefits from improved future prediction quality. The best results are **bolded**.

| Video model | Task completed in a row | | | | | Avg. Len. ↑ |
|---|---|---|---|---|---|---|
| | 1 | 2 | 3 | 4 | 5 | |
| SVD | 98.5 | 95.0 | 91.1 | 88.1 | 82.6 | 4.55 |
| Wan2.2 5B | **99.1** | **95.8** | **92.0** | **88.5** | **83.2** | **4.59** |

*Table 13.* **Comparing alternative IDM designs on CALVIN ABC-D.** We compare the proposed three-IDM decomposition against several single-branch alternatives. The best results are **bolded**.

| Methods | Task completed in a row | | | | | Avg. Len. ↑ |
|---|---|---|---|---|---|---|
| | 1 | 2 | 3 | 4 | 5 | |
| Direct multimodal inputs | 95.8 | 91.0 | 86.4 | 82.1 | 75.0 | 4.30 |
| Single VGGT IDM | 95.8 | 91.0 | 85.5 | 81.2 | 74.1 | 4.28 |
| Single RGB IDM | 96.2 | 91.1 | 86.5 | 82.6 | 74.9 | 4.31 |
| Single IDM w/ multi-modality loss | 96.5 | 91.0 | 87.5 | 82.4 | 77.4 | 4.35 |
| MoLA (three specialized IDMs) | **98.5** | **95.0** | **91.1** | **88.1** | **82.6** | **4.55** |

*Table 14.* **Inference-time controls for explicit future conditioning on CALVIN ABC-D.** MoIDM itself is preserved, while the task-conditioned imagined future is replaced by either an identical current frame or a noisy version of the current frame. The best results are **bolded**.

| Methods | Task completed in a row | | | | | Avg. Len. ↑ |
|---|---|---|---|---|---|---|
| | 1 | 2 | 3 | 4 | 5 | |
| Noisy-frame MoIDM | 94.0 | 88.9 | 83.3 | 77.4 | 70.7 | 4.14 |
| Same-frame MoIDM | 95.1 | 90.4 | 84.9 | 79.6 | 72.2 | 4.22 |
| MoLA | **98.5** | **95.0** | **91.1** | **88.1** | **82.6** | **4.55** |

limited ability to capture high-level spatial and semantic abstractions. Moreover, many existing VLA methods rely heavily on large-scale robot interaction data with ground-truth action annotations, which constrains scalability. In contrast, emerging works explore learning directly from internet-scale, action-free videos, enabling more scalable VLA training without explicit action supervision (Yang et al., 2025c; Zhou et al., 2025a; Luo et al., 2025).

Recently, extensive efforts have focused on pushing the upper limits of VLA capabilities from multiple perspectives, including incorporating 3D information (Qian et al., 2025; Jia et al., 2025b; Pan et al., 2025; Ze et al., 2024; Zhen et al., 2024b; Li et al., 2026a), integrating future prediction mechanisms (Zhao et al., 2025b; Cen et al., 2025a; Cai et al., 2026), leveraging reinforcement learning (Li et al., 2026c; Chen et al., 2025f; Lu et al., 2025a), enhancing reasoning abilities (Tan et al., 2026; Team et al., 2025; Wen et al., 2025a; Zhou et al., 2025c; Zhai et al., 2025), bridging the simulation-to-reality gap (Deng et al., 2025), improving instruction understanding (Li et al., 2025e;c), enabling cross-embodiment generalization (Zheng et al., 2025a; 2026), optimizing representations (Huang et al., 2025a; Lv et al., 2025; Chen et al., 2025b; Wen et al., 2024b; Zeng et al., 2024; Li et al., 2025d; Zhang et al., 2024c; Li et al., 2025a), supporting long-horizon manipulation (Bu et al., 2024b; Feng et al., 2025a; Zhang et al., 2025d), facilitating multi-system collaboration (Bu et al., 2024a; Xu et al., 2025; Chi et al., 2025a; Liu et al., 2025a; Bi et al., 2025), studying scaling laws (Hu et al., 2025b; Hou et al., 2025; Chen et al., 2025d), improving efficiency (Hung et al., 2025; Shukor et al., 2025), and modeling temporal information (Li et al., 2026b). In addition, some works investigate VLA deployment across more diverse robot embodiments (Wu et al., 2025b; Chen et al., 2025c; Liu et al., 2025b).

**Video-Action Models** World models learn internal representations of the environment (Chen et al., 2022; Robine et al., 2023; Wang et al., 2024; Peng et al., 2026) that enable agents to predict future states and make informed decisions. Recent advances have significantly improved the scalability and expressiveness of world models by adopting Transformer-based architectures capable of modeling long-horizon dynamics (Wu et al., 2025a; Robine et al., 2023; Micheli et al., 2023). These developments establish world models as a powerful paradigm for learning environment dynamics from visual experience.

Building upon world models, recent research has explored Video-Action models, which leverage video generation mod-

els (Agarwal et al., 2025; Wan et al., 2025; Blattmann et al., 2023; Chi et al., 2025b; Huang et al., 2026; Zhen et al., 2025; Gao et al., 2025; Lu et al., 2025b; Guo et al., 2025) as a core mechanism for robotic action prediction and control. In robotics, several works directly employ video generation models as policy backbones, decoding actions through tracking, inverse dynamics, or future-state matching (Black et al., 2023; Du et al., 2024; Yang et al., 2024b; Hu et al., 2025a; Liang et al., 2024; Liao et al., 2025; Tan et al., 2025a; Feng et al., 2025b; Huang et al., 2025b; Yang et al., 2025b; Routray et al., 2026; Shen et al., 2025; Zhou et al., 2025b; Jiang et al., 2025a; Fan et al., 2026; Sun et al., 2026). Methods such as UniPi (Du et al., 2024), DREAMGEN (Jang et al., 2025), and GeVRLM (Zhang et al., 2025b) generate future observations to guide action generation, while joint frameworks co-generate future frames and actions to improve temporal consistency and policy learning (Guo et al., 2024; Zheng et al., 2025b; Li et al., 2025b).

Future video prediction has also been widely adopted as a pretraining objective, as demonstrated in GR-2 (Cheang et al., 2024), VPP (Hu et al., 2025a), and RynnVLA-001 (Jiang et al., 2025b). Collectively, these approaches position Video-Action models as a unifying framework that bridges world modeling, video generation, and vision–language–action learning, highlighting the potential of predictive visual modeling to connect perception, dynamics, and action.

**Latent Action Models**    Latent action models aim to reduce reliance on dense action annotations by learning compact representations that capture environment dynamics and agent behaviors. Early approaches focus on learning variational latent spaces (Pu et al., 2016; Van Den Oord et al., 2017) from action trajectories, structuring action manifolds for behavior generation and task adaptation, as exemplified by VQ-BeT (Lee et al., 2024) and Quest (Mete et al., 2024). To exploit unlabeled videos, several works infer latent actions from visual dynamics by coupling inverse and forward dynamics models, learning latent variables that explain next-frame transitions (Rybkin et al., 2019; Edwards et al., 2019; Bruce et al., 2024; Bu et al., 2025a). Genie (Bruce et al., 2024) adopts a causal formulation to extract latent actions via next-frame prediction, while LAPO (Schmidt & Jiang, 2024) and DynaMo (Cui et al., 2024) learn latent actions directly from visual observations without explicit action supervision, particularly for manipulation tasks.

Recent methods such as LAPA (Ye et al., 2025) and IGOR (Chen et al., 2024) introduce unsupervised pretraining schemes for learning discrete latent actions within Vision–Language–Action models, enabling knowledge transfer from large-scale human videos. Most visual latent action models (Ma et al., 2025; Fan et al., 2025; Li et al., 2025f; Yang et al., 2025e) rely on RGB reconstruction, which inevitably captures task-irrelevant appearance variations (Zhang et al., 2025a). To mitigate this issue, subsequent works explore alternative supervision signals, including self-supervised visual features such as DINOv2 (Bu et al., 2025b; Chen et al., 2025e; Yang et al., 2025a), object-centric keypoints (Yang et al., 2025d; Collins et al., 2025; Yuan et al., 2025), and language-based objectives that provide semantic grounding (Clark et al., 2025). LAOM (Nikulin et al., 2025) further incorporates sparse action labels to bias latent representations toward controllable robotic behaviors.

**Future-Aware Control and Representation Learning**    Recent work on future-aware control spans both world action models and VLA-style future-aware representation learning. Within the world action model literature, VPP (Hu et al., 2025a), mimic-video (Pai et al., 2025), Genie Envisioner (Liao et al., 2025), and DiT4DiT (Ma et al., 2026) employ separate components to model future dynamics and action, whereas LingBot-VA (Li et al., 2026d) and Motus (Bi et al., 2025) couple the two more tightly through unified architectures. A parallel direction uses a single model to jointly generate future video and action, as exemplified by Unified Video Action Model (Li et al., 2025b), DreamZero (Ye et al., 2026b), GigaWorld-Policy (Ye et al., 2026a), and Cosmos Policy (Kim et al., 2026). Vidar (Feng et al., 2025b) and Large Video Planner (Chen et al., 2025a) first generate imagined visual futures and then convert them into executable actions through inverse-dynamics-style adaptation or retargeting. Large-scale pretraining for world action models is explored by Unified World Models (Zhu et al., 2025) and LDA-1B (Lyu et al., 2026), while Fast-WAM (Yuan et al., 2026) studies how to reduce test-time computation. In parallel, VLA-JEPA (Sun et al., 2026) and Joint-Aligned Latent Action (Luo et al., 2026) employ future-aware objectives primarily for representation shaping or training-time supervision within VLA frameworks. Relative to these directions, MoLA is most closely related to world action models that retain an explicit future-imagination module, while differing in its use of multiple pretrained IDMs as a latent-action interface between imagined video and control.

### A.5. Discussion

**Architectural Role of Latent Actions.**    Within this design space, methods also differ in how latent actions participate in downstream control. For example, ViPRA (Routray et al., 2026) combines large-scale robot-video pretraining with latent actions, but its latent action tokenization primarily serves as an auxiliary signal for shaping the representations learned by

a large video-language model during pretraining. By contrast, MoLA uses an explicit IDM-style latent-action interface during downstream training and inference: imagined futures are first compressed into latent actions, and these latent actions are then consumed by the action head to generate executable controls. Accordingly, MoLA is better characterized as a world-model-plus-action-head architecture with a dedicated latent-action interface, rather than a VLM-centric route.

## A.6. Qualitative Results

**Video Generation Results** Figure 6 shows qualitative results of our video generation model for both the primary and wrist camera views. The model generates coherent and visually plausible future frames, accurately capturing object movements, interactions, and robot motions. These high-quality predictions provide a reliable foundation for the subsequent MoLA, ensuring that the inferred latent actions are grounded in realistic imagined futures. Overall, the results demonstrate that our video prediction model effectively supports action-centric reasoning for robot manipulation.

**Qualitative Analysis of One-Step Denoising.** Figure 7 provides a qualitative illustration of the one-step denoising regime used in MoLA. Even with a single denoising step, the video model preserves the temporal structure most relevant to control and still provides MoIDM with reliable transition cues for extracting action-relevant latent actions. In MoLA, the video generation module is designed to provide a compact and task-relevant future hypothesis rather than a photorealistic video prediction. A small number of denoising steps can therefore be advantageous, as it may suppress redundant texture and appearance details and encourage the model to focus on control-critical dynamics.

**MoIDM Reconstruction Results.** We visualize the reconstruction quality of MoIDM on RGB frames, depth maps, and optical flow to assess its ability to capture modality-specific information. As shown in Figure 8, each modality-specific IDM produces highly accurate reconstructions, preserving both spatial structures and motion dynamics. These results indicate that MoIDM effectively encodes essential features across multiple modalities, supporting consistent and coherent spatiotemporal representations.

**MoIDM Attention Heatmaps.** We visualize the attention heatmaps of MoIDM to examine its capability to capture action-related information. Figure 9 illustrates that across different datasets in both simulated and real-world environments, the model consistently focuses its attention on the robot arm region at each time step. This observation further indicates that MoIDM contributes to improving the model's understanding of actions and facilitates effective alignment between action semantics and actual execution.

**Benchmarks.** We visualize the experimental results on the CALVIN, LIBERO, and LIBERO-Plus benchmarks. As shown in Figure 10, the results demonstrate that our method achieves strong performance across all benchmarks, accurately understanding and following given instructions to complete diverse tasks in a coherent and stable manner.

**Failure Cases.** We further conduct a qualitative analysis of several failure cases through visualization. As illustrated in Figure 11, since our method generates action sequences via video prediction based on a world model, it exhibits certain limitations in reasoning and error correction under complex scenarios, compared to approaches that rely on explicit reasoning and policy generation with large language models. In most cases, the model is able to correctly understand and execute the given instructions; however, in rare situations involving unexpected state changes (e.g., a mug tipping over) or execution deviations (e.g., grasping an incorrect object), the model struggles to perform timely self-correction, which may lead to task failure. These observations indicate that there remains room for improvement in enhancing robustness to anomalous situations and error correction capabilities.

**Sensitivity to Video Generation Quality.** Figure 12 presents two representative cases for analyzing sensitivity to video generation quality. In the first case, the generated future deviates from the task-relevant evolution and leads to task failure. In the second case, the generated video contains action prediction errors, yet the policy still succeeds. These examples show that video generation quality affects downstream execution, but not in an absolute manner. Systematic errors in imagined futures can bias the latent actions extracted by MoIDM and consequently mislead the action head. At the same time, MoLA remains relatively robust because it does not map generated video to control at the pixel level. Instead, MoIDM compresses future visual transitions into action-centric latent actions, from which the action head can still recover reasonable actions by exploiting preserved dynamical cues under imperfect conditional representations.

**Real-World Qualitative Analysis.** We provide qualitative results to evaluate the real-world generalization ability of our method. Figure 13 illustrates the training and evaluation settings used in our real-world experiments. As shown in Figure 14, under in-distribution conditions, the model is able to accurately interpret instructions and consistently execute tasks in a stable and coherent manner, demonstrating reliable real-world performance. More importantly, our method exhibits strong generalization capability under out-of-distribution settings. As illustrated in Figures 15 and 16, the model successfully handles novel object appearances, unseen object configurations, and variations in the environment that are not observed during training. Despite these distribution shifts, the model is able to generate reasonable action sequences and complete tasks without additional fine-tuning. These qualitative results indicate that our approach learns robust action representations that transfer effectively to real-world scenarios, highlighting its strong generalization ability beyond the training distribution.

**High-Precision Continuous Control.** We further evaluate MoLA on a more complex real-world task that serves as a more direct test of fine manipulation. This setting requires tighter pose control, stronger contact stability, and more consistent temporal coordination, making it substantially more demanding than the other real-world tasks considered in this work. As illustrated in Figure 17, MoLA achieves a 45% success rate on this task, demonstrating that the proposed action-centric interface remains effective under challenging continuous control settings.

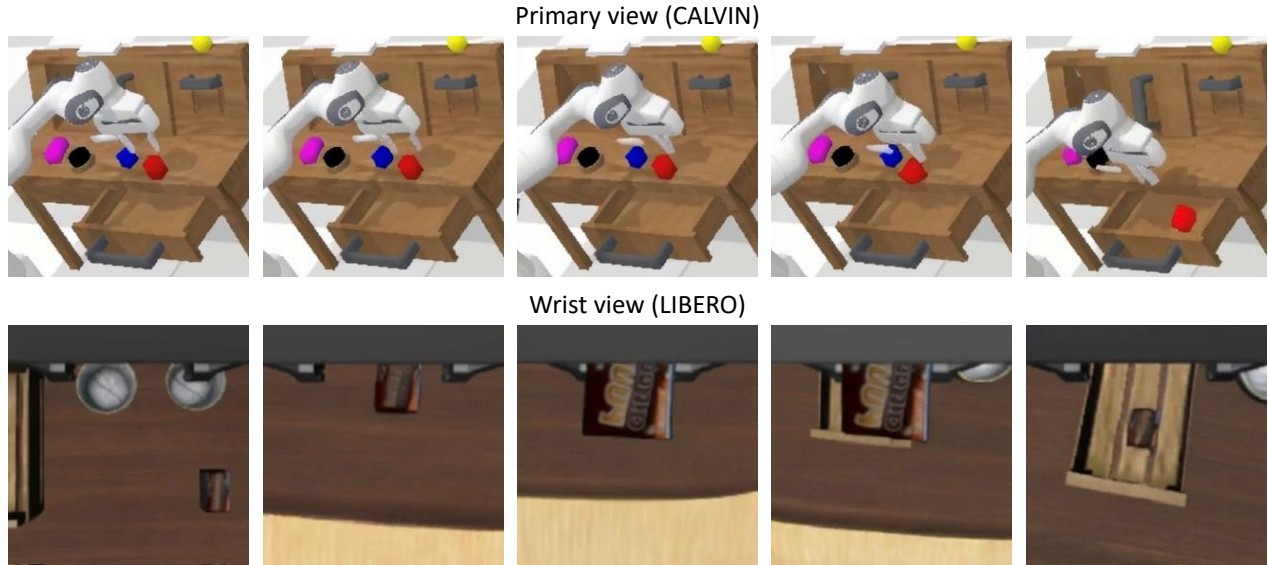

*Figure 6.* **Qualitative results** of the video generation model on primary and wrist views.

### Open the middle drawer of the cabinet

*Figure 7.* **Qualitative analysis of one-step denoising.** A single denoising step preserves control-relevant temporal structure and provides MoIDM with reliable transition cues, even without producing a fully photorealistic future video.

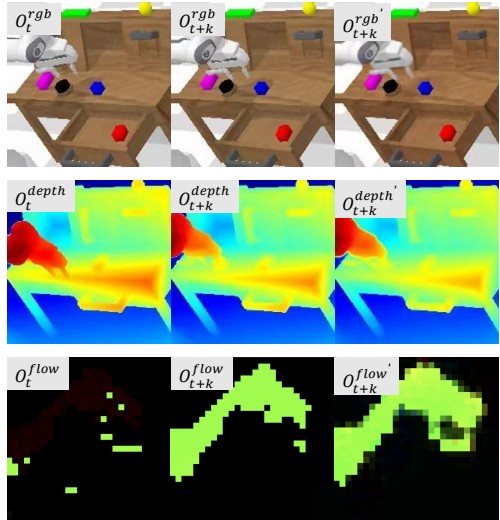

*Figure 8.* **Reconstruction results of MoIDM.** The semantic-aware IDM performs feature-level reconstruction without visualization.

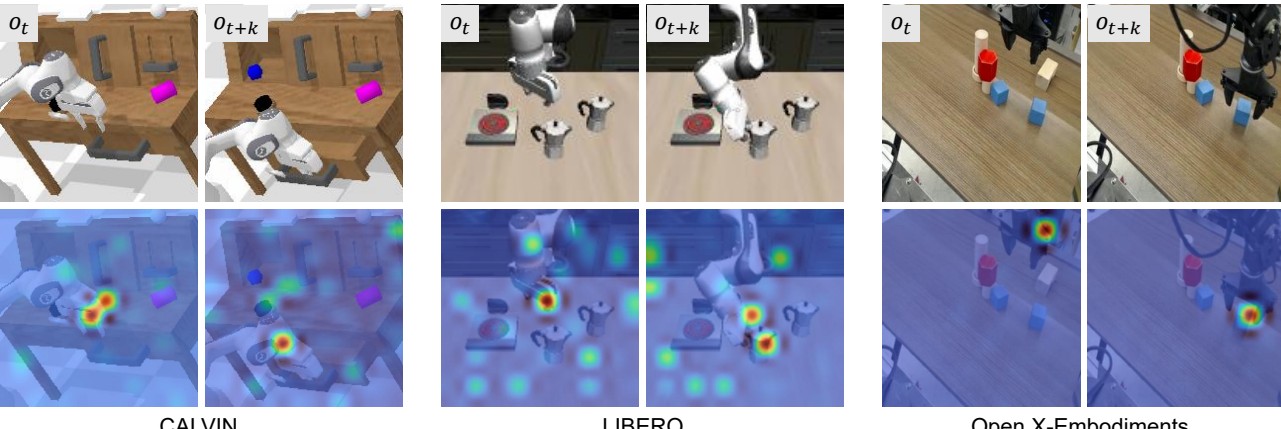

*Figure 9.* **Visualization results of MoIDM attention heatmaps.** The attention consistently localizes on action-critical regions.

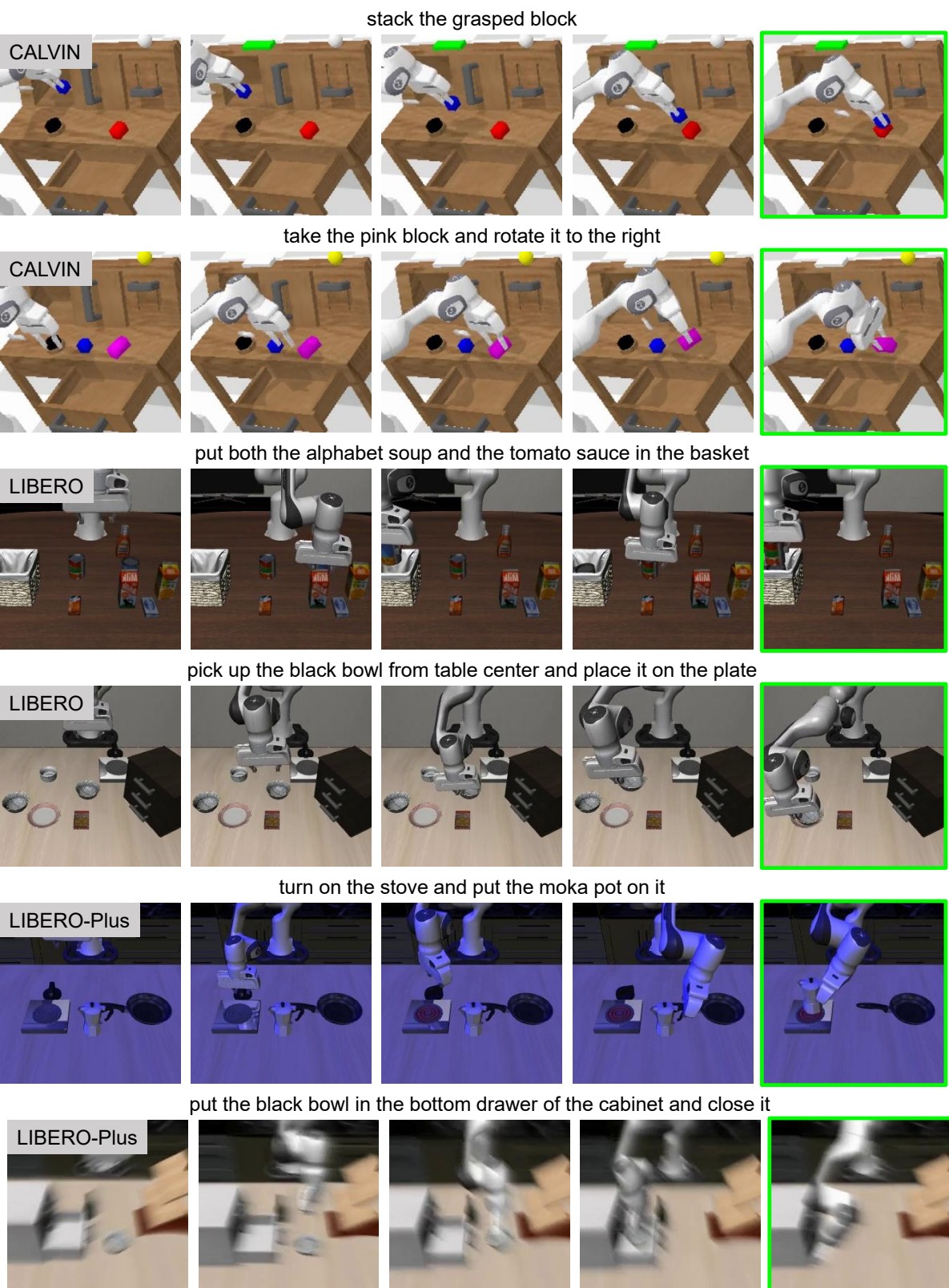

*Figure 10.* **Qualitative results** on the CALVIN, LIBERO, and LIBERO-Plus benchmarks.

put the yellow and white mug in the microwave and close it

pick up the black bowl on the ramekin and place it on the plate

*Figure 11.* **Qualitative results** of failure cases.

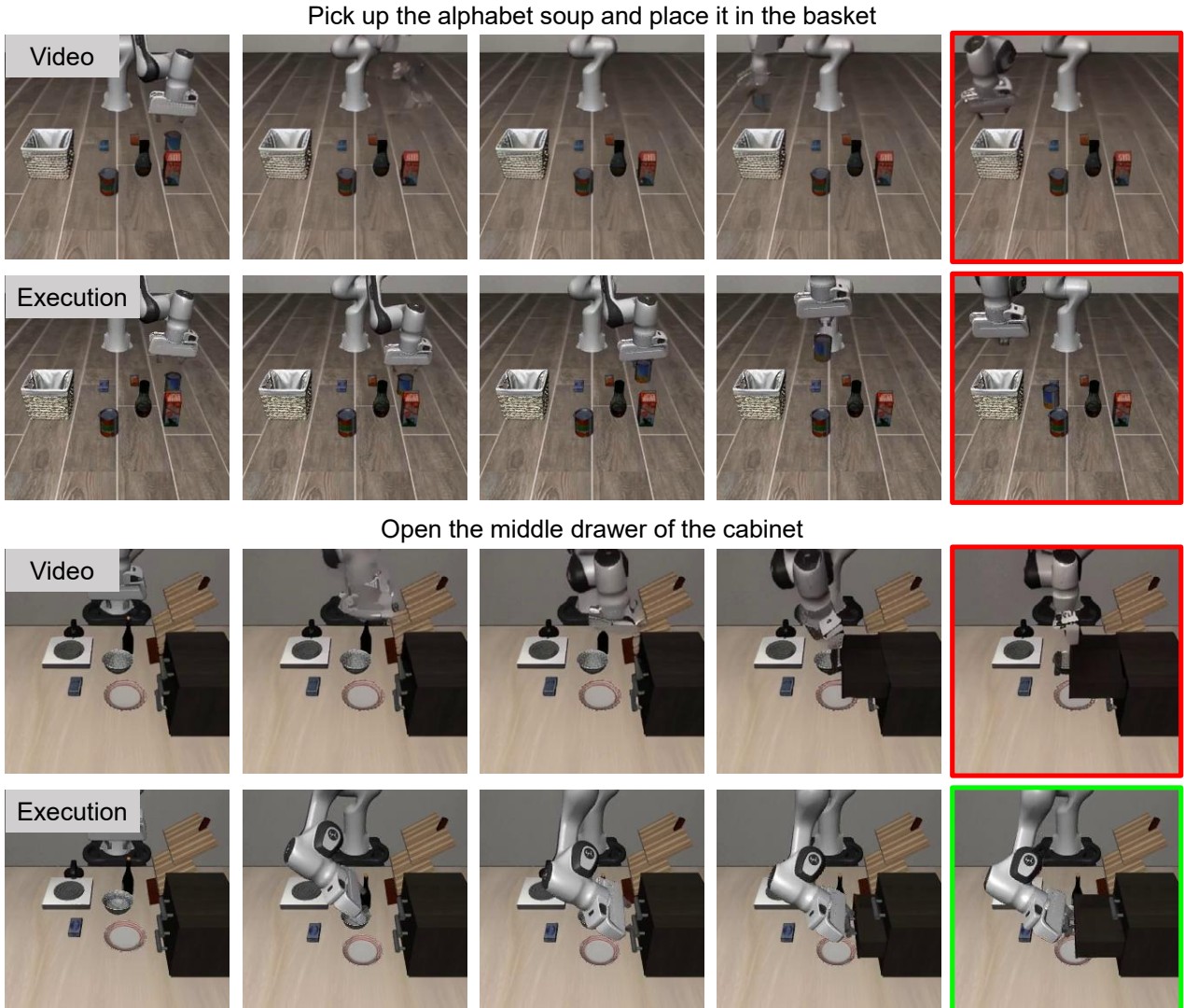

*Figure 12.* **Sensitivity of MoLA to video generation quality.** In one case, the generated future deviates from the task-relevant evolution and causes failure; in the other, the policy still succeeds despite action prediction errors in the generated video.

(a) Training

| *place the blue bottle into the bin* | *place the yellow bottle into the bin* | *place the purple bottle into the bin* | *grab the bowl to the side* | *grab the bowl into the box* |

(b) In-distribution evaluation

(c) Out-of-distribution evaluation

*Distracting objects*

*Changes in lighting conditions*

*Figure 13.* **Training and evaluation settings** for our real-world experiments.

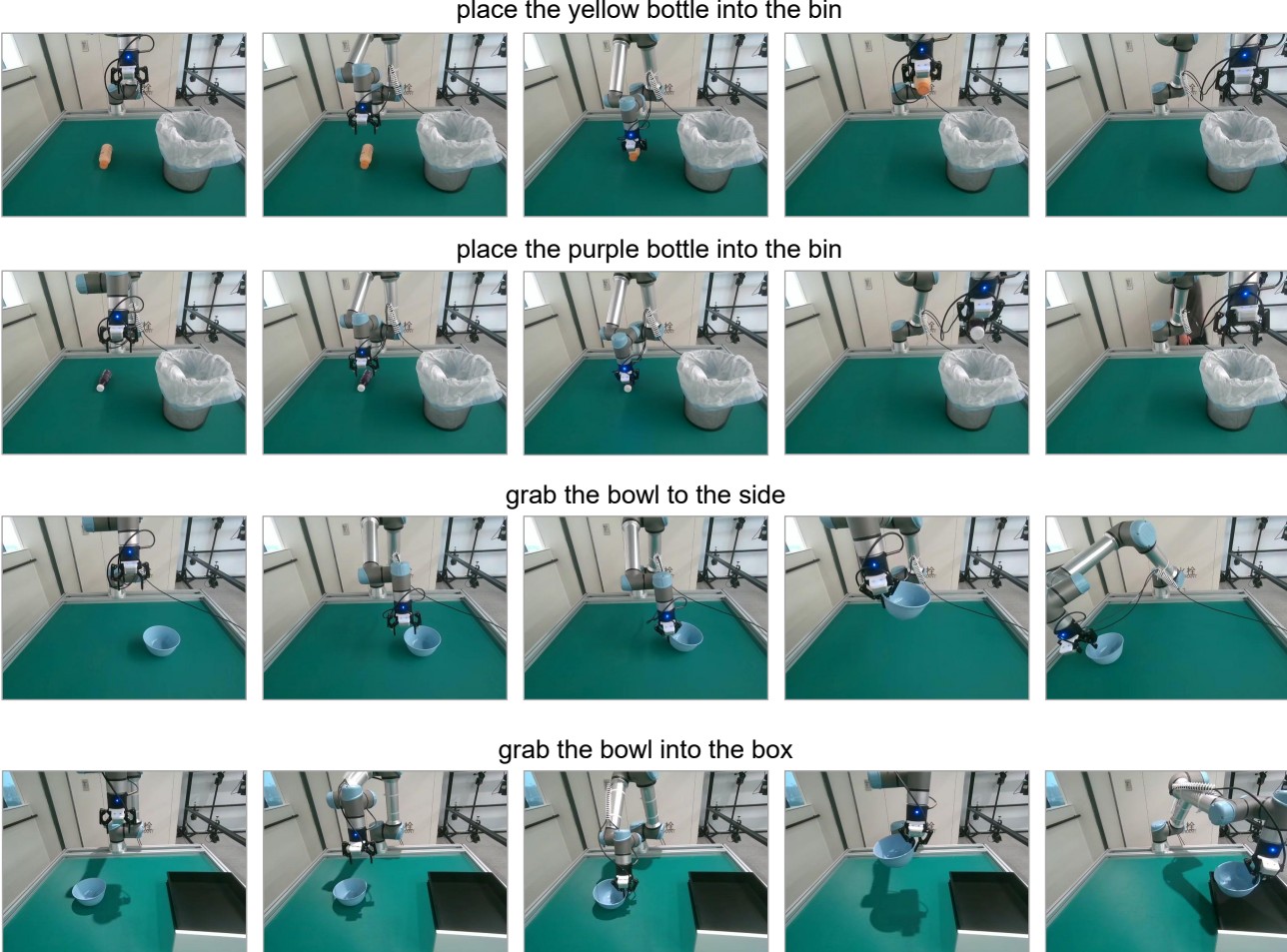

*Figure 14.* **Qualitative results** of in-distribution real-world evaluation.

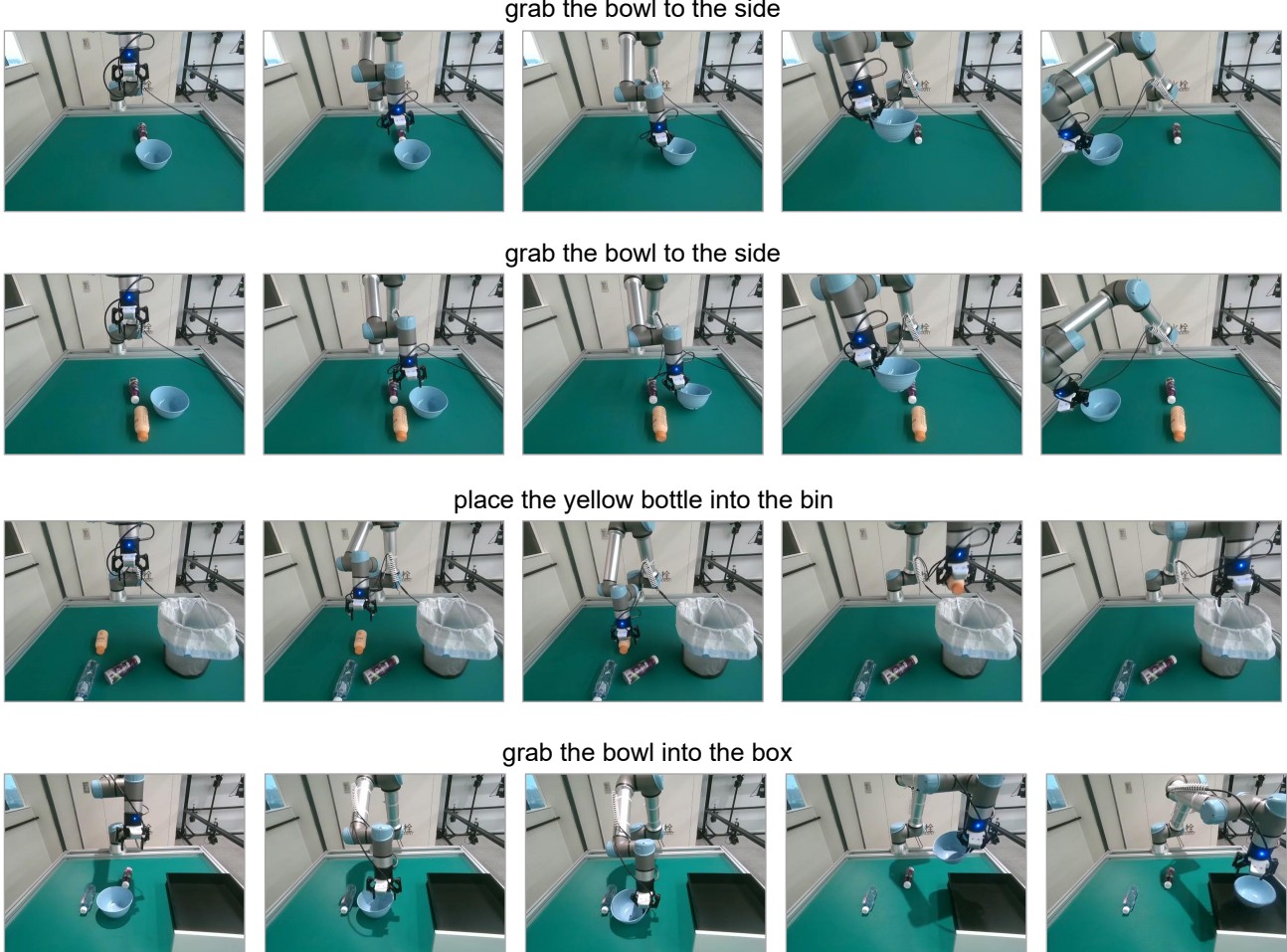

*Distracting objects*

*Figure 15.* **Qualitative results** of out-of-distribution real-world evaluation.

place the yellow bottle into the bin

place the purple bottle into the bin

place the blue bottle into the bin

place the yellow bottle into the bin

*Changes in lighting conditions*

*Figure 16.* **Qualitative results** of out-of-distribution real-world evaluation.

pour water from a teapot into a glass

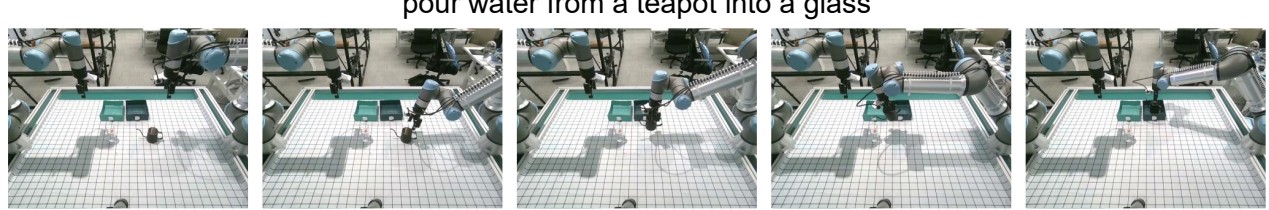

*Figure 17.* **High-precision continuous control on a more complex real-world task.** The task requires tighter pose control, stronger contact stability, and more consistent temporal coordination.

