# OpenReview forum: "From Imagined Futures to Executable Actions: Mixture of Latent Actions for Robot Manipulation"
_ICML.cc/2026/Conference — ICML 2026 regular_

### Official Review · Reviewer_d4ja · 2026-03-08

**Soundness:** 2
**Presentation:** 2
**Significance:** 3
**Originality:** 3
**Overall Recommendation:** 4
**Confidence:** 4

**Summary:**

This paper is motivated by a challenge in imagination-based robotics: video generation models produce visually realistic but not action-oriented future predictions. Rather than feeding predicted frames directly to a policy, MoLA passes them through three specialized latent action models, each sensitive to a different modality (optical flow, depth, and semantics), to extract informative latent action representations. These modality-aware latent action models are pretrained on large-scale robot datasets, and then jointly fine-tuned end-to-end with a diffusion-based action head. Experiments across CALVIN, LIBERO, LIBERO-Plus simulations, and a real UR5e robot show consistent improvements over prior video-action and VLA baselines.

**Compliance With Llm Reviewing Policy:**

Affirmed.

**Final Justification:**

The authors addressed my experimental concerns in the rebuttal. Although the writing contains several vague descriptions of the motivation and contributions, after several rounds of discussion, I agree that these issues will not critically affect the paper's overall evaluation. I believe they can be fully resolved in the camera-ready version.

**Key Questions For Authors:**

- Methods like LAPA also pretrain on large-scale video datasets to learn latent actions, without explicit video prediction. What is the advantage of MoLA over such approaches?
- In Table 4, all of MoLA's results are bolded even when they are not the best, while bolding in other tables denotes the best result. This inconsistency is misleading.
- The paper frequently uses "inverse dynamics models" to refer to the models that predict latent actions. However, existing literature [1-2] commonly uses IDM to denote models that predict final executable actions. This might cause some confusion.

[1] Baker, Bowen, et al. "Video pretraining (vpt): Learning to act by watching unlabeled online videos." Advances in Neural Information Processing Systems 35 (2022): 24639-24654.

[2] Du, Yilun, et al. "Learning universal policies via text-guided video generation." Advances in neural information processing systems 36 (2023): 9156-9172.

**Limitations:**

yes

**Strengths And Weaknesses:**

Strengths:
- The paper proposes a novel approach that introduces more structured supervision signals into the video pretraining process.
- Experiments are conducted across three simulation benchmarks and a real-world setup, with comparisons against multiple policy learning methods, VLA models, and Video-Action models, where MoLA achieves strong performance.

Weaknesses: Despite MoLA's strong performance, the key takeaways from the experiments remain unclear. As I understand, MoLA introduces two components over existing video-action models like VPP: (1) compressing predicted frames into a lower-dimensional yet informative latent action representation as an input to the policy head, and (2) incorporating more structured visual signals like depth, segmentation, and motion flow. Comparing VPP in Table 1 against the "Baseline" in Table 5 suggests that component (1) alone does not outperform VPP. This raises the question: whether the gains are driven primarily by component (2), or whether the authors intend to argue that (1)+(2) is a necessary combination. Would directly feeding depth, segmentation, and motion flow as inputs to the policy head achieve comparable performance?

---

> ### Author Rebuttal · Authors · 2026-03-30
>
> We sincerely thank the reviewer for the thoughtful and constructive feedback; below we provide clarifications and additional comparisons.
>
> **W1:** What mainly drives MoLA’s gains: latent-action compression, structured visual signals, or only their combination? Would directly feeding depth, semantic, and flow features to the policy head perform similarly?
>
> **W1:** We will clarify in the revision that the baseline in Table 5 uses neither three IDMs nor a VQ-VAE-style discrete latent-action bottleneck, but instead aggregates future visual features from the video model with a temporal Transformer and feeds them directly into the action head. All additional experiments below are on CALVIN ABC-D: Baseline removes all IDMs and uses only Transformer aggregation; Direct multimodal inputs feed depth, semantic, and motion flow features directly into the action head; and MoLA w/ RGB-only IDM keeps only the RGB-reconstruction-supervised latent-action-compression IDM. The results show that directly feeding structured visual signals still leaves the action head to align modalities, extract action-relevant causality, and decode control from high-dimensional perceptual features, whereas MoLA first organizes these cues into a control-aligned latent action space. Thus, the gain comes from the combination of structured visual signals and latent-action compression, not either alone. We will include these results and analysis in the revision.
>
> | Methods | 1 | 2 | 3 | 4 | 5 | Avg. Len. |
> | --- | --- | --- | --- | --- | --- | --- |
> | Baseline | 95.9 | 90.4 | 85.1 | 80.0 | 72.8 | 4.24 |
> | Direct multimodal inputs | 95.8 | 91.0 | 86.4 | 82.1 | 75.0 | 4.30 |
> | MoLA w/ RGB-only IDM | 96.2 | 91.1 | 86.5 | 82.6 | 74.9 | 4.31 |
> | MoLA (Ours) | 98.5 | 95.0 | 91.1 | 88.1 | 82.6 | 4.55 |
>
> **Q1:** What advantage does MoLA have over latent-action methods like LAPA that do not use explicit video prediction?
>
> **Q1:** The key difference from latent-action methods such as LAPA is that LAPA learns latent action tokens and predicts them directly from the current observation, whereas MoLA first imagines task-relevant futures with a world model and then uses MIDM to convert future visual transitions into action-centric latent actions. Because MIDM is pretrained on large-scale robot data, it injects action priors rather than learning the latent extractor only from limited downstream data. This future-driven intermediate representation reduces the mismatch between observations and actions, since latent actions come from explicit future state changes rather than being guessed from a single frame. It also better exploits temporal dynamics and interaction priors from the video model, which should improve generalization to unseen scenes, appearance changes, and distribution shifts.
>
> **Q2:** MoLA seems worse than $\pi_{0.5}$ on some real tasks; Table 4 formatting is misleading.
>
> **Q2:** We are sorry for the unclear presentation. We will clarify in the revision that the bolding in Table 4 is not intended to imply a strict apples-to-apples win over $\pi_{0.5}$, which is an extremely strong but scope-misaligned reference: a company-scale, heavily pretrained end-to-end VLA built with large-scale heterogeneous co-training, rather than the world model + action-centric interface studied here. For MoLA, more direct mechanistic comparators are world-model-based baselines such as VPP. We include $\pi_{0.5}$ as a strong VLA reference point, not to claim that MoLA fully outperforms the strongest traditional VLA, and we will make this explicit in both the caption and main text to avoid misunderstanding.
>
> **Q3:** The term “inverse dynamics model” seems misused here, since it usually refers to predicting executable actions rather than latent actions.
>
> **Q3:** We are sorry for the imprecise wording. In the classical literature, IDM usually denotes a model that directly predicts executable actions, whereas our method predicts latent actions that are later decoded by the action head into continuous control. Calling it an IDM can therefore be confusing; more precisely, it is a latent action model that serves an inverse-dynamics role rather than a traditional executable-action IDM. In the revision, we will make the terminology consistent throughout the paper, clarify that these modules output latent actions, and align more closely with prior usage of IDM to avoid ambiguity.

---

> > ### Author Rebuttal · Reviewer_d4ja · 2026-04-04
> >
> > Thank you for the rebuttal. The clarifications for Q1/Q2/Q3 and the additional experiments for W1 are helpful. However, I still have several concerns about the paper's writing and framing:
> > - The paper's central narrative is that predicted futures are not inherently action-centric, and that MoLA addresses this by transforming imagined future videos into action-centric latent representations, rather than feeding predicted frames directly to the policy. However, the additional ablations in the rebuttal support a narrower contribution: neither latent-action compression alone nor structured multimodal signals alone appears sufficient, and the gains seem to come primarily from their combination. This creates a mismatch between the paper's central narrative and the more coupled empirical contribution supported by the rebuttal.
> > - The introduction claim that "existing uses ... do not provide inverse dynamics models pretrained on large-scale robot datasets that can robustly operate on generated futures" may be factually inaccurate. If IDM is interpreted in the latent-action sense, then related work such as ViPRA (which is also cited in the paper) already combines large-scale robot-video pretraining with future-video and latent-action prediction, even if its architecture is not identical to MoLA.
> > - The issue around IDM is not merely a matter of terminology inconsistency. Statements such as "This gap motivates the use of inverse dynamics models, which predict the action that causes a transition between observations" literally describe classical executable-action IDMs, not latent-action modules. Inaccurate statements of this kind make it difficult to understand the intended scope of the authors' claims.
> >
> > I appreciate the paper's technical contribution. However, I believe the writing would require substantial revision, so my score remains unchanged.
> >
> > **Update (after reading the reply rebuttal comment):**
> > *(I just realized that new comments are not visible to authors, so I’m adding this update here.)*
> >
> > Thanks for your clarification. Let me restate the concerns more clearly.
> >
> > (2) Regarding the introduction claim: The sentence and the nearby context saying that prior work does not provide "inverse dynamics models pretrained on large-scale robot datasets that can robustly operate on generated futures" seem to invite a new paradigm: large-scale robotic pretraining for models that predict future visual observations and infer action-relevant latent representations from current and future frames. Although ViPRA is described as "jointly predict future visual observations and latent actions within a VLM", its prediction factorization is still sequential (i.e., latent action decoder is still conditioned on current and future visual tokens). In that sense, I think ViPRA falls within the same paradigm. Therefore, my concern is whether the claim as written may overstate the novelty.
> >
> > (1) Regarding the central narrative: Across the abstract and introduction, the paper seems to frame the core problem as predicted futures being insufficiently action-centric, and the core solution as converting them into compressed latent-action representations before policy prediction. Semantic, depth, and flow cues are certainly discussed, but they are mostly presented as complementary signals that strengthen this latent-action interface, rather than as co-equal drivers of the contribution. Although the additional experiment for W1 shows some improvement for MoLA w/ RGB-only IDM, I do not think the gain is strong enough to support latent-action compression as the paper's primary contribution. Moreover, as I noted in the above concern (2), it is also unclear whether latent-action compression itself should be regarded as a strictly new contribution. For these reasons, I still feel like the paper's central narrative might need be adjusted accordingly, to better reflect that the strongest empirical support is for the combination of latent-action compression and structured multimodal supervision.
> >
> > (3) Using this terminology is okay, but more importantly, statements about the paper's motivation and contribution should make the intended referent clear. I appreciate the authors' willingness to revise this point.
> >
> > If the authors can convince me on concerns 1 and 2, I would be open to reconsidering my score.

---

> > > ### Author Response · Authors · 2026-04-04
> > >
> > > We sincerely thank the reviewer for reading the paper so carefully. However, we believe **some of the remaining concerns stem from a misunderstanding of the paper**.
> > >
> > > **1.** Our experiments show that both latent-action compression and structured multimodal signals are independently useful, and that **their combination yields a distinctly stronger joint gain**.
> > >
> > > - Latent-action compression is supported by the gain over the baseline from the RGB-supervised latent-action compression variant and by the gap between direct multimodal inputs and full MoLA (see our response to W1).
> > >
> > > - Structured multimodal signals are supported by the improvement of direct multimodal inputs over the baseline (see our response to W1) and by Reviewer BCNQ's W1, where even a single latent-action branch benefits from multimodal supervision.
> > >
> > > In this sense, **the full set of results actually broadens, rather than narrows, the paper's contribution**.
> > >
> > > **2.** **It seems there is a misunderstanding**. In our paper, IDM was used for modules that **predict latent actions from image pairs of current and future frames**. ViPRA's core design is to train a video-language model to **jointly predict future visual observations and motion-centric latent actions**.
> > >
> > > **3.** We appreciate the reviewer for noting that at least one place in our paper uses IDM in the classical executable-action sense, and we will revise this more carefully. At the same time, influential works such as LAPA [1] and UniVLA [2] also use inverse-dynamics language without explicitly separating executable and latent actions. In this sense, **our wording followed existing usage**.
> > >
> > > [1] Latent Action Pretraining from Videos, ICLR 2025.
> > >
> > > [2] Learning to Act Anywhere with Task-centric Latent Actions, RSS 2025.
> > >
> > > We hope this clarification helps the reviewer and the AC assess **the paper's actual contribution** more accurately.
> > >
> > > ---
> > >
> > > **Updated on April 6, 2026**
> > >
> > > **We highly value this kind of exchange**, which is aimed at improving the quality of the paper. Below, we provide more specific clarifications.
> > >
> > > **(2)** In prior work, IDM is commonly understood as a module that extracts action representations from pairs of current and future frames. Here, latent actions are not merely abstract variables for representation learning; rather, they are **action-rich representations that can be mapped to executable actions** by a simple action head.
> > >
> > > - Although ViPRA also pretrains a Latent Action Tokenizer, its primary role is to serve **as an auxiliary signal during pretraining to supervise the learning of a large-scale video-language model**. More precisely, during fine-tuning and inference, ViPRA relies more on the predictive strength of the pretrained VLM itself.
> > >
> > > - Moreover, the sentence right before that claim states: "**imagined future frames** can be converted into executable representations." Therefore, the context here is specifically about a **video-model-based** route, whereas ViPRA is closer to a **VLM-based** route. They are not exactly the same technical object that this sentence is intended to delimit.
> > >
> > > We are very grateful for your careful examination of the boundary of our claim and for the thoughtful comparison between our work and these strong prior works. **We will make this relationship clearer in the revised manuscript**.
> > >
> > > **(1)** In our view, the core design of MoLA is that **it is among the earlier "world model + action head" approaches to use an explicit IDM-style interface for imagined futures**, which is different from the more common "VLM backbone + action head" route.
> > >
> > > - Latent-action compression is **the core interface design that connects imagined futures with executable control**, consistent with our response to Q1. On top of this, the role of structured multimodal signals should be understood more accurately as making this paradigm more physically grounded.
> > >
> > > - **MoLA's contribution is to propose and validate a clear technical path**. In our experiments, RGB-only latent-action compression is **already effective**. From a broader technological perspective, we do not rule out the possibility that pure RGB latent-action compression may achieve even stronger performance in future work. However, that is not the core question this paper seeks to answer. Rather, this paper aims to **provide an initial yet concrete empirical starting point for this direction, demonstrating the feasibility of the route itself**.
> > >
> > > Accordingly, we agree that the revised manuscript should make the hierarchy among these core paradigms clearer, but **we do not believe this reduces the paper to a narrower contribution**.
> > >
> > > Finally, **we again sincerely thank you for such constructive engagement with the paper and will revise the manuscript carefully based on it**. At the same time, we would also like to emphasize that these revisions are mainly final polishing at the writing level, and are **fully feasible to complete in the final version**.

---

### Official Review · Reviewer_HfH2 · 2026-03-08

**Soundness:** 3
**Presentation:** 3
**Significance:** 2
**Originality:** 2
**Overall Recommendation:** 4
**Confidence:** 4

**Summary:**

This paper introduces MoLA (Mixture of Latent Actions), a framework designed to resolve the disconnect between visually realistic video generation and action-centric control in Vision-Language-Action models. Instead of mapping predicted video frames directly to continuous actions, MoLA employs Stable Video Diffusion to generate future visual rollouts , which are then processed by multiple pretrained inverse dynamics models (MIDM). These models extract a discrete mixture of latent actions that capture complementary semantic, depth, and flow cues. This structured representation, alongside the generated visual features, is used to condition a Diffusion Transformer (DiT) action head that is optimized via a flow matching objective to output executable robot control commands. The method is evaluated on simulated benchmarks like CALVIN and LIBERO, as well as on a real-world UR5e robot, and shows improvement over several VLA and video-action model baselines.

**Compliance With Llm Reviewing Policy:**

Affirmed.

**Final Justification:**

The authors have addressed all my concerns through additional experiments, and I believe the paper offers valuable insights and improvement upon a popular architecture useful for robotics. I will thereby raise my score to a weak accept.

**Key Questions For Authors:**

Ordered based on importance:
- Given that latent actions are discretized via VQ codebooks, what advantage does the diffusion-based DiT action head provide over a simpler autoregressive transformer? An ablation study can be helpful to justify design choice.
- The video model uses only one denoising step at inference. How sensitive is performance to the number of denoising steps?
- Report inference time for all experiments, especially real world; currently it is only reported for the CALVIN benchmark.
- How does the approach perform when paired with stronger video generation backbones?
- [Presentation] Please add a y-axis label in Figure 4. Add sim benchmark name in the caption for Table 5 (I assume it's CALVIN ABC-D).

**Limitations:**

Limitations are discussed in the appendix, yet the authors are encouraged to include more discussions in terms of training complexity, video quality vs. inference time trade-offs, and real-world failure cases.

**Strengths And Weaknesses:**

**Strengths:**
- The framework cleverly bridges the visual-to-action gap by using multiple inverse dynamics models (MIDM) to infer latent actions, and the chosen combo of IDMs achieve the best performance.
- It leverages frozen foundation models—specifically Depth Anything v2, SAM2, and CoTracker3—to supervise the extraction of complementary geometric, semantic, and motion flow cues.
- Evaluation shows better performance against strong baselines such as Open-VLA or VPP, with additional evaluation on real world robot arm.

**Weaknesses:**
- Deploying a Diffusion Transformer (DiT) optimized via flow matching for the action head introduces architectural complexity , yet the paper fails to ablate whether this continuous diffusion process is actually necessary for decoding action latents that have already been discretized into tokens via vector-quantized codebooks.
- To achieve viable inference speeds, the Stable Video Diffusion module is severely restricted to a single denoising step, forcing a compromise between the visual fidelity of the imagined future and execution latency.
- The physical evaluations on the UR5e robot are restricted to relatively simple, short-horizon tasks, such as placing a bottle into a bin or grasping a bowl. This makes it difficult to assess the system's ability to handle dexterous, high-precision tasks.
- The approach relies on Stable Video Diffusion, but it is unclear whether the proposed framework generalizes to stronger or more recent DiT-based video generation models such as Wan 2.2.

---

> ### Author Rebuttal · Authors · 2026-03-30
>
> We appreciate the reviewer's thoughtful comments and suggestions, and below address the main questions with further clarification.
>
> **W1:** Is the continuous diffusion in the DiT action head necessary, given that latents are already tokenized via VQ codebooks?
>
> **W1&Q1:** The discrete latent actions are not final controls, but intermediate cues distilled from imagined futures. The policy must still model a continuous, temporally correlated, and often multimodal action distribution, for which diffusion- and flow-based policies have proven effective (e.g., Diffusion Policy [1] and $\pi_0$ [2]). We therefore use a DiT + flow matching head for final action generation. As a lighter ablation, we also test an autoregressive Transformer over discretized action tokens. It works, confirming that the latent actions are informative, but is clearly weaker than the DiT-based head, which better captures long-horizon dependencies, multimodality, and stable control under imperfect latent cues. We will include this ablation in the revision as direct support for this design choice.
>
> | Action head | 1 | 2 | 3 | 4 | 5 | Avg. Len. |
> | --- | --- | --- | --- | --- | --- | --- |
> | AR token Transformer | 95.7 | 91.4 | 88.6 | 85.1 | 78.9 | 4.40 |
> | DiT + flow matching | 98.5 | 95.0 | 91.1 | 88.1 | 82.6 | 4.55 |
>
> [1] Diffusion Policy Visuomotor Policy Learning via Action Diffusion, RSS 2023.
>
> [2] $\pi_0$: A Vision-Language-Action Flow Model for General Robot Control, RSS 2025.
>
> **W2:** Does limiting Stable Video Diffusion to a single denoising step create a trade-off between imagined future quality and execution latency?
>
> **W2&Q2:** We will add visualizations of one-step imagined futures in the revision (available at [the anonymous link](https://anonymous.4open.science/r/icml26_HfH2)). Even with a single denoising step, the video model preserves the temporal structure most relevant to control; although not high-fidelity, it still provides MIDM with reliable transition cues for extracting action-relevant latent actions. In MoLA, the video module is meant to provide a compact, task-relevant future hypothesis rather than a photorealistic video. Fewer denoising steps may even suppress redundant texture and appearance details, helping the model focus on control-critical dynamics.
>
> **W3:** Do the UR5e evaluations generalize to more dexterous, high-precision tasks?
>
> **W3:** We apologize that the current UR5e experiments mainly assess deployability and out-of-distribution generalization rather than high-precision continuous control. To address this, we add a more complex real-robot task following the main-text setting (available at [the anonymous link](https://anonymous.4open.science/r/icml26_HfH2)), on which MoLA achieves a 45% success rate. Due to rebuttal time limits, this is the only additional task validated so far. It is a more direct test of fine manipulation, requiring tighter pose control, contact stability, and temporal consistency, and will be included in the revision as an additional qualitative result.
>
> **W4:** Does the framework generalize to stronger or newer DiT-based video generators like Wan 2.2?
>
> **W4&Q4:** MoLA is not tied to SVD and can naturally adopt stronger DiT-based video generators. To verify this, we add results using Wan2.2 5B as the imagination backbone on CALVIN ABC-D, showing that MoLA is compatible with stronger video backbones and can benefit from improvements in the underlying imagination model.
>
> | Video Model | 1 | 2 | 3 | 4 | 5 | Avg. Len. |
> | --- | --- | --- | --- | --- | --- | --- |
> | SVD | 98.5 | 95.0 | 91.1 | 88.1 | 82.6 | 4.55 |
> | Wan2.2 5B | 99.1 | 95.8 | 92.0 | 88.5 | 83.2 | 4.59 |
>
> **Q1:** What advantage does the diffusion-based DiT action head have over a simpler autoregressive transformer?
>
> **Q1**: See **W1**.
>
> **Q2:** The video model uses only one denoising step at inference. How sensitive is performance to the number of denoising steps?
>
> **Q2**: See **W2**.
>
> **Q3:** Report inference time for experiments, especially real world.
>
> **Q3:** We reported module-level inference times on CALVIN ABC-D in the paper. The other experiments have highly similar inference cost because they use the same model configuration and 256x256 inputs. We will clarify this in the revision and add a real-world inference-time table to make deployment cost more explicit.
>
> | Model part | Real-world inference time |
> | --- | --- |
> | Video generation model | 81.23 ms |
> | Multiple inverse dynamics models | 11.05 ms |
> | Action head | 9.62 ms |
>
> **Q4:** How does the approach perform when paired with stronger video generation backbones?
>
> **Q4:** See **W4**.
>
> **Q5:** Add a y-axis label in Figure 4 and specify the simulation benchmark in Table 5 caption.
>
> **Q5:** We apologize for these omissions. In the revision, we will add the missing y-axis label in Figure 4 and specify the benchmark in the Table 5 caption. The updated figure is available at [the anonymous link](https://anonymous.4open.science/r/icml26_HfH2).

---

> > ### Author Rebuttal · Reviewer_HfH2 · 2026-04-02
> >
> > Thanks the authors for the additional experiments and clear rebuttal, most of my concerns are addressed. While the current results suggest that one-step video prediction already provides useful signals for downstream policy learning, it would be valuable to more systematically characterize this trade-off. In particular, I recommend including a table that reports these metrics in each column:
> >
> > - Number of video diffusion inference steps
> > - Policy performance (e.g., success rate and/or action MSE)
> > - Total Inference time
> >
> > Happy to raise my score to a weak accept if this is adequately addressed.

---

> > > ### Author Response · Authors · 2026-04-03
> > >
> > > We sincerely thank the reviewer for the encouraging follow-up comments and the very positive reassessment of our work. We also appreciate the suggestion to characterize this trade-off more systematically, which would further strengthen the paper. Below, we provide the requested analysis on CALVIN ABC-D and will include it in the revised manuscript. All experiments measuring inference speed are conducted on an NVIDIA GeForce RTX 4090 GPU, with timing averaged over ten runs.
> > >
> > > | Number of video diffusion inference steps | Avg. Len. | Total inference time |
> > > | --- | --- | --- |
> > > | 1 | 4.55 | 0.1296 s |
> > > | 2 | 4.52 | 0.2431 s |
> > > | 10 | 4.56 | 1.1659 s |
> > > | 20 | 4.49 | 2.3185 s |
> > >
> > > These results suggest that one-step denoising already provides highly effective control-relevant signals while offering by far the lowest inference cost. Increasing the number of denoising steps does not lead to a monotonic policy improvement, which is consistent with our design goal: MoLA does not require photorealistic future video, but rather compact future hypotheses that preserve the temporal structure most useful for action inference.
> > >
> > > We are again very thankful for this thoughtful follow-up question and for the reviewer's generous consideration. We will make sure this analysis is clearly included in the revised manuscript.

---

### Official Review · Reviewer_3PB6 · 2026-03-12

**Soundness:** 2
**Presentation:** 3
**Significance:** 2
**Originality:** 3
**Overall Recommendation:** 4
**Confidence:** 3

**Summary:**

This paper proposes a MoLA (Mixture of Latent Actions) structure to leverage multiple inverse dynamics models to learn modality-specific action-centric latents, which helps the transformation from generated videos to robot actions. MoLA adopts multiple inverse dynamics model pretraining and end to end finetuning. Experiments show the strength of the proposed MoLA structure in simulation benchmarks and real world experiments. Ablation studies help to understand the design of the model structure and the pretraining - finetuning pipeline.

**Compliance With Llm Reviewing Policy:**

Affirmed.

**Final Justification:**

Overall, the paper is well motivated by incorporating multi-modality IDMs for robot manipulation, and the additional experiments during rebuttal verify the importance of multi-IDM design. That said, some real-world performance comparisons are still mixed, and the sensitivity to generated video quality is still a limitation. For these reasons, I revise my recommendation to be weak accept.

**Key Questions For Authors:**

1. It seems that the proposed MoLA performs worse than $\pi$-0.5 in some real world tasks. However, Table 4 is not bolded as it should be, which is quite misleading.
2. What does "baseline" in table 5 mean? Maybe one important ablation is to use only one LDM but with multiple modality loss. Will one strong LDM better than 3 MLDMs?
3. How does the model perform compared with RGB-based inverse dynamics model?
4. Is the performance sensitive to the video generation quality?

**Limitations:**

What are the failure cases for this model? Will the video generation quality affect the robot execution a lot?

**Strengths And Weaknesses:**

Strengths:

This paper bridges the gap between video generation imagination and real robot actions through modality-specific latent inverse dynamics models. Experiments show the improvement compared to related models, and ablation helps the understanding of model structure and training schedule.

Weakness:

1. The improvement of the multiple inverse dynamics model is not significant, and maybe need an ablation study for 1 LDM with multiple modality supervision loss compared with multiple LDMs.
2. This paper's central claim is that MLDM helps inverse dynamics, however, the experiments mainly compare with other VLAs. Some comparison to RGB-only inverse dynamics module will help the understanding.

---

> ### Author Rebuttal · Authors · 2026-03-30
>
> We sincerely thank the reviewer for the thoughtful and constructive feedback; below we provide additional ablations, clarifications, and discussion to address the main concerns.
>
> **W1:** Improvement of multi-IDM unclear; ablation: single vs multiple IDMs?
>
> **W1:** Regarding the comparison between "a single stronger multi-modality IDM" and "three modality-specific IDMs," we fully agree that this is an important ablation. The additional comparison results are as follows:
>
> | Methods | 1 | 2 | 3 | 4 | 5 | Avg. Len. |
> | --- | --- | --- | --- | --- | --- | --- |
> | Single IDM w/ multi-modality loss | 96.5 | 91.0 | 87.5 | 82.4 | 77.4 | 4.35 |
> | MoLA (Ours) | 98.5 | 95.0 | 91.1 | 88.1 | 82.6 | 4.55 |
>
> A single shared IDM forces all modalities into one codebook and latent space, increasing objective interference and representation trade-offs. In contrast, three specialized IDMs preserve complementary inductive biases, which the action head fuses into more robust, interpretable action-centric latents. Our design therefore intentionally factorizes control-relevant visual information into 3D structure, motion flow, and appearance/semantic context, rather than arbitrarily stacking branches.
>
> **W2:** Some comparison to RGB-only inverse dynamics module will help the understanding.
>
> **W2&Q3:** We will add a more direct comparison experiment in the revision, comparing the full MoLA and our own implementation of MoLA w/ RGB-only IDM on CALVIN ABC-D. The results are as follows:
>
> | Methods | 1 | 2 | 3 | 4 | 5 | Avg. Len. |
> | --- | --- | --- | --- | --- | --- | --- |
> | MoLA w/ RGB-only IDM | 96.2 | 91.1 | 86.5 | 82.6 | 74.9 | 4.31 |
> | MoLA (Ours) | 98.5 | 95.0 | 91.1 | 88.1 | 82.6 | 4.55 |
>
> Although a single RGB IDM preserves texture and appearance, it entangles multiple factors in one representation, making it more sensitive to appearance noise and control-irrelevant variation. MIDM is designed to prevent RGB-only latent action learning from absorbing such irrelevant factors: structured depth, dynamics, and semantic supervision aligns latent actions more closely with executable control. By contrast, an IDM trained only with RGB reconstruction lacks explicit motion constraints and is therefore more vulnerable to visual artifacts in imagined futures.
>
> **Q1:** MoLA seems worse than $\pi_{0.5}$ on some real tasks; Table 4 formatting is misleading.
>
> **Q1:** We apologize that our presentation in Table 4 was not sufficiently clear. We will clarify in the revision that the bolding in Table 4 is not intended to imply a strict apples-to-apples win over $\pi_{0.5}$, which is an extremely strong but scope-misaligned reference: a company-scale, heavily pretrained end-to-end VLA built with large-scale heterogeneous co-training, rather than the world model + action-centric interface studied here. For MoLA, more direct mechanistic comparators are world-model-based baselines such as VPP. We include $\pi_{0.5}$ as a strong VLA reference point, not to claim that MoLA fully outperforms the strongest traditional VLA, and we will make this explicit in both the caption and main text to avoid misunderstanding.
>
> **Q2:** Table 5 “baseline” meaning? Single strong LDM vs three MLDMs?
>
> **Q2:** We will clarify in the revision that the Table 5 baseline uses neither three IDMs nor a VQ-VAE-style discrete latent-action bottleneck. Instead, it aggregates future visual features predicted by the video model with a temporal Transformer and feeds the resulting conditional representation directly into the action head. Its performance further suggests that without IDM, the action head must learn the observation-to-action mapping directly from aggregated future visual features, substantially increasing the difficulty of control modeling. The corresponding ablation and analysis are provided in **W1**.
>
> **Q3:** How does the model perform compared with RGB-based inverse dynamics model?
>
> **Q3:** See **W2**.
>
> **Q4&Limitations:** What are the failure cases for this model? Is the performance sensitive to the video generation quality?
>
> **Q4&Limitations:** We provide two representative cases (available at [the anonymous link](https://anonymous.4open.science/r/icml2026_3PB6)): in one, video generation deviates from the task-relevant future and causes failure; in the other, despite action prediction errors in the generated video, the model still succeeds. These cases show that video generation quality affects execution, but not absolutely: systematic errors in imagined futures can bias the latent actions extracted by MIDM and thus mislead the action head, yet MoLA remains robust because it does not map generated video to control at the pixel level. Instead, MIDM compresses future visual transitions into action-centric latent actions, from which the action head can still recover reasonable actions using preserved dynamical cues and robust modeling under imperfect conditional representations. We will include these visualized examples in the revision to make this point clearer.

---

> > ### Author Rebuttal · Reviewer_3PB6 · 2026-04-04
> >
> > Thanks for the author's rebuttal. I think my concerns have been fully resolved. The ablation with a single model but with multi-modality loss, and the ablation with RGB-only IDM, will help to understand why the modality-specific IDM design is important. Thus, I will increase my score to weak accept.

---

> > > ### Author Response · Authors · 2026-04-04
> > >
> > > Dear Reviewer,
> > >
> > > We greatly appreciate your positive response and the time you devoted to reviewing our work. We are very glad that our rebuttal and the additional ablation studies have adequately addressed your concerns.  Your encouragement means a great deal to us, and we will continue polishing the manuscript accordingly.
> > >
> > > Wishing you all the best.
> > >
> > > Warm regards,
> > >
> > > Authors

---

### Official Review · Reviewer_BCNQ · 2026-03-13

**Soundness:** 3
**Presentation:** 3
**Significance:** 3
**Originality:** 2
**Overall Recommendation:** 5
**Confidence:** 4

**Summary:**

This paper studies how to make imagined futures more actionable for robot manipulation. The key argument is that predicted future frames are visually informative but not directly optimized for control, so a better interface is needed between future imagination and executable actions. To address this, the paper proposes MoLA, which first generates imagined future rollouts, then applies multiple modality-aware inverse dynamics models to translate future transitions into a mixture of latent actions, and finally predicts executable actions with a diffusion-based action head. The modality-aware branches are specialized through semantic, depth, and flow supervision. The system is trained in stages, with the video generator adapted first, the inverse dynamics modules pretrained next, and the final downstream training performed by freezing the video generator while jointly optimizing the inverse dynamics modules and the action head. Empirically, the method achieves strong performance on CALVIN, LIBERO, LIBERO-Plus, and real-world manipulation tasks.

Overall, I found the paper interesting, well-motivated, and empirically strong. The central idea is intuitive and useful: instead of consuming imagined future frames directly, the policy benefits from a more control-oriented latent action interface derived through inverse dynamics. My main reservations are that the design remains somewhat heuristic and that the paper should narrow the scope of its claims. In particular, the current evidence supports MoLA as an effective IDM-based latent-action interface for imagined futures, but does not yet fully establish whether explicit online video prediction at inference time is fundamentally necessary.

**Compliance With Llm Reviewing Policy:**

Affirmed.

**Final Justification:**

I found this paper strong already in the original submission, and the rebuttal further increased my confidence in it. The core contribution is to provide an action-centric interface between imagined future rollouts and executable control, rather than feeding predicted future frames directly into the policy. In my view, this is the right level at which to understand the paper’s contribution, and it is both intuitively meaningful and empirically well supported by the results across CALVIN, LIBERO, LIBERO-Plus, and real-world manipulation.

What I found especially valuable in the rebuttal is that it added mechanism-level evidence rather than only clarification. The new ablations on unified versus multiple IDMs, RGB-only variants, direct multimodal inputs, and controls that weaken or remove meaningful future-transition cues all help clarify what is driving the gains. Taken together, these results support a coherent picture: neither raw future visual features alone nor a weaker latent interface alone is sufficient, while the proposed action-centric interface becomes substantially stronger when paired with the structured multimodal supervision introduced in the paper.

I also paid attention to the remaining concerns from other reviewers about framing and positioning. I agree that the final version should calibrate these points more carefully, especially in how it presents the relationship among latent-action compression, multimodal supervision, and nearby prior work. However, I view this primarily as a matter of sharpening the paper’s claims and exposition, rather than a reduction of its substantive contribution. The rebuttal makes it sufficiently clear that the paper’s strongest supported claim is not an overly broad one, but a concrete and meaningful technical one: MoLA provides an effective interface for converting imagined futures into action-centric control representations, and the proposed multimodal design materially strengthens this route.

Overall, I believe this is a technically strong paper with a clear insight, broad evaluation, meaningful real-world validation, and a rebuttal that substantially improved the mechanism-level credibility of the work. I therefore raise my overall recommendation to 5.

**Key Questions For Authors:**

1. Can the paper better separate the role of online imagined futures from the role of latent action learning?

The most important mechanism-level question is whether the gains come from test-time imagined rollouts themselves, or from the latent action representation induced by training with imagined futures.

A particularly valuable follow-up would be to take the IDM-based latent action learning paradigm proposed here and combine it with a two-stage latent-action pipeline. Concretely, imagined futures could be used during training to shape the latent action space through the proposed MIDM mechanism, while the downstream policy would then predict these latent actions directly at inference time, without requiring the video generator online. This is not simply a request to compare against a different method as a black box; rather, it would directly test the core mechanism behind MoLA. It would clarify whether explicit online imagination is necessary or whether most of the benefit comes from the learned latent action abstraction, which could potentially yield a much lighter inference pipeline.

2. Why is this particular modality decomposition the right one?

The results suggest that semantic, depth, and flow supervision are complementary, which is encouraging. Still, I would appreciate a clearer discussion of why these three branches in particular are the preferred factorization, and whether the authors considered alternatives such as a unified inverse dynamics model or other future-oriented supervisory signals.

3. Can the paper discuss its position in the broader future-aware control landscape a bit more explicitly?

I do not view this as a weakness of the current submission, especially given how quickly this area is evolving. Still, the revised or final version would benefit from a brief discussion of some very recent concurrent directions on future-aware latent action learning and training-time future injection. This would help better situate MoLA within the broader design space of explicit imagined rollouts, latent action interfaces, and future-aware representation shaping.

DreamZero: World Action Models are Zero-shot Policies
VLA-JEPA: Enhancing Vision-Language-Action Model with Latent World Model
Joint-Aligned Latent Action: Towards Scalable VLA Pretraining in the Wild

**Limitations:**

One important limitation is that the method remains relatively heavy at inference time because it still relies on online video generation. The paper shows that this can work well, but it does not yet establish whether this cost is fundamentally necessary.

**Strengths And Weaknesses:**

## Strengths
 - The paper identifies an important bottleneck in imagination-based manipulation: visually plausible future predictions do not automatically translate into control-relevant representations. Framing the problem as the need for an intermediate, action-centric interface is, in my view, the strongest conceptual contribution of the paper.
- The proposed solution is intuitive. Using inverse dynamics to explain future transitions is a natural design choice, and the semantic/depth/flow factorization is a reasonable way to capture complementary future-aware cues.
- The empirical evaluation is strong and fairly broad. The method performs well on CALVIN, LIBERO, LIBERO-Plus, and real-robot tasks, so the evidence is not confined to a single benchmark.
- The ablations are useful. The paper shows that the different modality-aware branches are complementary, that MIDM pretraining is important, and that jointly adapting MIDM with the action head is beneficial.


## Weaknesses

1. The design still feels somewhat heuristic

    While the method is sensible and effective, it also has a fairly engineered flavor: imagined futures from a video generator, several specialized inverse dynamics modules, and a downstream action head. The paper demonstrates that semantic, depth, and flow constraints are complementary, but it does not fully establish why this specific decomposition is the right one, rather than one effective design among several plausible alternatives. Similarly, it remains unclear whether multiple specialized inverse dynamics branches are truly necessary or whether a stronger unified latent-action model could recover much of the same benefit.

2. It remains unclear whether online video prediction at inference time is necessary

    This is my main reservation. MoLA still depends on online imagined rollouts at test time. Even if the video generator is frozen and simplified, this makes the overall inference path heavier than some alternative future-aware control designs. More importantly, the current experiments do not fully disentangle whether the gains come from explicit online imagined futures or from the latent action abstraction learned under future-based supervision. As a result, the paper convincingly shows that MoLA is an effective system, but it does not yet establish the stronger conclusion that test-time future generation is itself essential.

3. The scope of the claims should be slightly narrowed

    The strongest supported claim, in my view, is that MoLA is an effective IDM-based latent-action interface for leveraging imagined futures in manipulation. That is already a meaningful contribution. I would encourage the paper to present itself in that more precise scope, rather than implying a broader conclusion about the necessity of explicitly imagined rollouts for future-aware control.

---

> ### Author Rebuttal · Authors · 2026-03-31
>
> We sincerely thank the reviewer for the constructive and insightful feedback. Our detailed responses are provided below.
>
> **W1:** While semantic, depth, and flow constraints complement each other, it is unclear why this decomposition is optimal or if multiple inverse dynamics branches are necessary versus a unified latent-action model.
>
> **W1&Q2:** We apologize for not clearly motivating this design in the original submission. To directly justify the three-IDM design, we will add the following comparison in the revision:
>
> | Methods | 1 | 2 | 3 | 4 | 5 | Avg. Len. |
> | --- | --- | --- | --- | --- | --- | --- |
> | Single VGGT IDM | 95.8 | 91.0 | 85.5 | 81.2 | 74.1 | 4.28 |
> | Single RGB IDM | 96.2 | 91.1 | 86.5 | 82.6 | 74.9 | 4.31 |
> | Single IDM w/ multi-modality loss | 96.5 | 91.0 | 87.5 | 82.4 | 77.4 | 4.35 |
> | MoLA | 98.5 | 95.0 | 91.1 | 88.1 | 82.6 | 4.55 |
>
> VGGT mainly provides geometric/spatial priors, without explicit functional decomposition; a single RGB IDM entangles multiple factors and is more sensitive to control-irrelevant appearance variation; and a shared multi-modal IDM forces all signals into one codebook and latent space, increasing objective interference and representation trade-offs. In contrast, MoLA's three specialized IDMs preserve complementary inductive biases that the action head fuses into more robust, interpretable action-centric latents.
>
> **W2:** It is unclear if test-time video prediction is needed; it is uncertain whether gains come from latent action abstraction or online future generation.
>
> **W2&Q1:** Thank you for this insightful suggestion. We agree that explicit online imagined futures and future-supervised latent action abstraction should be separated more clearly. In MoLA, MIDM itself does not predict futures; it takes the current frame and an imagined future and extracts action-centric latent actions from their transition. The proposed two-stage setting is therefore a valuable control for testing how much performance remains without explicit test-time imagination. We add two inference-time controls: feeding MIDM either two identical current frames or the current frame plus a noisy version of itself. Both preserve MIDM while removing the task-conditioned future transition cues from the world model. These results show that MoLA relies on both latent action learning and explicit future cues during inference.
>
> | Methods | 1 | 2 | 3 | 4 | 5 | Avg. Len. |
> | --- | --- | --- | --- | --- | --- | --- |
> | Noisy-frame MIDM | 94.0 | 88.9 | 83.3 | 77.4 | 70.7 | 4.14 |
> | Same-frame MIDM | 95.1 | 90.4 | 84.9 | 79.6 | 72.2 | 4.22 |
> | MoLA | 98.5 | 95.0 | 91.1 | 88.1 | 82.6 | 4.55 |
>
> **W3:** The paper should focus on MoLA as an effective IDM-based latent-action interface, not on the necessity of explicit imagined rollouts.
>
> **W3:** We apologize that our previous framing was not sufficiently precise. In the revision, we will narrow the claim and position MoLA more explicitly as a latent-action interface within imagined-based control. A variant without online future hypotheses is an insightful, lighter follow-up, but it belongs to a different method class and is orthogonal to our core claim. MoLA's core contribution is using MIDM to convert imagined futures into action-centric representations that make imagined-based control more stable and effective.
>
> **Q1:** Can the paper better separate online imagined futures from latent action learning? A useful follow-up is to predict latent actions at inference without an online video generator.
>
> **Q1:** See **W2**.
>
> **Q2:** Why choose this modality decomposition? It would help to clarify why these three branches are preferred over alternatives such as a unified inverse dynamics model or other future-oriented signals.
>
> **Q2:** See **W1**.
>
> **Q3:** Can the paper better position itself in the future-aware control landscape? A brief discussion of recent work would help situate MoLA.
>
> **Q3:** We apologize for the unclear positioning. MoLA's contribution is a latent-action interface for leveraging future information in imagined-based manipulation. Like DreamZero, it is a world action model, but DreamZero jointly models future video and action, whereas MoLA uses pretrained MIDM to convert generated video into action-centric latents decoded into continuous control. By contrast, VLA-JEPA and Joint-Aligned Latent Action are closer to injecting future-aware signals into VLA models. We therefore position MoLA as an interface design within world action models, and we have added more comparisons with recent work at [the anonymous link](https://anonymous.4open.science/r/icml26_BCNQ) for inclusion in the revised manuscript.
>
> **Limitations:** A key limitation is that inference is still heavy due to reliance on online video generation.
>
> **Limitations:** To limit the overhead of online video generation, we use only a single denoising step at inference; together with the reported inference-time statistics, this keeps MoLA's overall cost within an acceptable range.

---

> > ### Author Rebuttal · Reviewer_BCNQ · 2026-04-01
> >
> > The rebuttal has convincingly addressed my main concerns. In particular, the introduction of a unified IDM variant and the additional ablations disentangling prediction from IDM significantly strengthen the paper. These experiments directly respond to my questions about whether multiple specialized IDMs are necessary and whether the gains stem from latent action abstraction or test-time future prediction.
> > Overall, these clarifications substantially improve the credibility of the proposed method and provide much clearer insight into the underlying mechanisms. I find the updated empirical evidence both thorough and well-aligned with the key conceptual claims. As a result, my concerns have been largely resolved.

---

> > > ### Author Response · Authors · 2026-04-02
> > >
> > > Dear Reviewer,
> > >
> > > Thank you again for your thoughtful comments and constructive suggestions. Your feedback has been very valuable and has greatly helped us improve the quality and clarity of our manuscript. We truly appreciate your positive evaluation and encouraging words, which motivate us to further revise and strengthen the paper.
> > >
> > > We sincerely wish you success in your research and all the best in your work.
> > >
> > > Warm regards,
> > >
> > > The Authors

---

### Decision · Program_Chairs · 2026-04-30

**Decision:**

Accept (regular)

**Comment:**

This paper received mixed initial ratings but the rebuttal successfully addressed the primary concerns with all reviewers recommending (weak or clear) accept. The AC supports the consensus to accept and urges the authors to improve the presentation as promised (e.g. the IDM terminology) as well as include the additional ablations from the rebuttal in the main text.